# Decoupled transcript and protein concentrations ensure histone homeostasis in different nutrients

Dimitra Chatzitheodoridou, Daniela Bureik, Francesco Padovani 🆔, Kalyan V Nadimpalli & Kurt M Schmoller 🆔 ✉

## Abstract

**To maintain protein homeostasis in changing nutrient environments, cells must precisely control the amount of their proteins, despite the accompanying changes in cell growth and biosynthetic capacity. As nutrients are major regulators of cell cycle length and progression, a particular challenge arises for the nutrient-dependent regulation of 'cell cycle genes', which are periodically expressed during the cell cycle. One important example are histones, which are needed at a constant histone-to-DNA stoichiometry. Here we show that budding yeast achieves histone homeostasis in different nutrients through a decoupling of transcript and protein abundance. We find that cells downregulate histone transcripts in poor nutrients to avoid toxic histone overexpression, but produce constant amounts of histone proteins through nutrient-specific regulation of translation efficiency. Our findings suggest that this allows cells to balance the need for rapid histone production under fast growth conditions with the tight regulation required to avoid toxic overexpression in poor nutrients.**

**Keywords** Budding Yeast; Histones; Protein Homeostasis; Nutrients; Cell Cycle
**Subject Categories** Cell Cycle; Chromatin, Transcription & Genomics

## Introduction

Protein homeostasis is critical for cell viability and function. Precise regulation of protein amounts must therefore be ensured despite changes in cell morphology, growth, and cell cycle progression that are caused by cellular programs such as cell differentiation, or external factors such as nutrient availability, temperature, oxygen concentration, and osmotic pressure (Jorgensen and Tyers, 2004; Watanabe and Okada, 1967; Ortmann et al, 2014; Taïeb et al, 2021; Smets et al, 2010; Broach, 2012; Pérez-Hidalgo and Moreno, 2016).

For example, in a nutrient-rich medium, cells typically grow larger and faster than in a nutrient-poor medium (Broach, 2012; Brauer et al, 2008; Johnston et al, 1979; Korem Kohanim et al, 2018;

Sauls et al, 2019). At the same time, rich nutrients lead to elevated ribosome concentrations, which facilitates higher protein synthesis rates and increased biosynthesis (Waldron and Lacroute, 1975; Metzl-Raz et al, 2017). Nutrients are also major regulators of cell cycle progression and passage through the cell size checkpoints (Bohnsack and Hirschi, 2004; Kim et al, 2002). Limitation of nutrients or growth factors can induce cell cycle arrest or lengthening of the G1-phase (Jorgensen and Tyers, 2004; Foster et al, 2010). For example, budding yeast cells growing slowly on poor medium usually spend more time in G1 until they reach the appropriate size to pass *Start*, the commitment point to cell cycle entry (Johnston et al, 1977; Parviz and Heideman, 1998; Qu et al, 2019). As a result, the durations of S and G2/M-phase become relatively shorter (Leitao and Kellogg, 2017). In the context of protein homeostasis, such nutrient-induced changes in cell cycle length and progression pose a challenge for the large fraction of "cell cycle genes", which are periodically expressed in a cell cycle-dependent manner (Fischer et al, 2022; Whitfield et al, 2002; Ishida et al, 2001; Dolatabadi et al, 2017; Spellman et al, 1998). To maintain these proteins at constant concentrations across nutrient conditions, cells would need to adjust protein synthesis rates to changes in cell cycle progression. However, whether and how cells ensure homeostasis of cell cycle-regulated proteins in changing environments is still unclear.

Histones are a prime example of proteins which are produced in a strongly cell cycle-dependent manner and whose concentration needs to be precisely controlled. To couple histone amounts to the genome content and coordinate histone synthesis with DNA replication, the expression of replication-dependent core histone genes is initiated in the late G1-phase and then continues throughout the S-phase (Eriksson et al, 2012; Duronio and Marzluff, 2017; Mendiratta et al, 2019; Marzluff and Duronio, 2002). Moreover, in contrast to most proteins, histone production is independent of cell volume, which ensures that histone amounts are tightly coupled to the genome content, even though total protein amounts increase with cell volume (Wiśniewski et al, 2014; Claude et al, 2021; Swaffer et al, 2021; Lanz et al, 2022). In budding yeast, this cell volume-independent production of histones is already achieved on a transcript level (Claude et al, 2021; Swaffer et al, 2021), at least in part through regulation by the promoter (Claude et al, 2021). This raises the question of whether similar regulation also ensures histone homeostasis in changing environments.

Institute of Functional Epigenetics, Molecular Targets and Therapeutics Center, Helmholtz Zentrum München, 85764 Neuherberg, Germany.
✉E-mail: kurt.schmoller@helmholtz-munich.de

Here, we use the budding yeast *Saccharomyces cerevisiae* as a model to understand how cells produce correct amounts of histones in different nutritional environments, despite the accompanying changes in cell size, growth rate, and cell cycle fractions. We find that while cells maintain constant stoichiometry between DNA and histone proteins, histone transcript concentrations are paradoxically decreased in poor compared to rich nutrients—potentially to avoid histone overexpression, which is more toxic in poor growth conditions. This decoupling of histone mRNA and protein amounts across different nutrients implies the requirement for nutrient-dependent regulation of histone translation efficiency. By combining single-cell approaches with population-level analysis, we were able to show that histone promoters are sufficient to mediate nutrient-dependent transcript regulation, and its compensation by translation.

## Results

### Histone protein concentrations decrease with cell volume across nutrient conditions

To determine the impact of nutrient availability on the regulation of histone protein concentrations, we grew wild-type haploid cells on synthetic complete (SC), yeast peptone (YP), and minimal medium (SD), containing glucose (D) or galactose (Gal) as fermentable and glycerol and ethanol (GE) as non-fermentable carbon sources. The selected set of seven different growth media resulted in a wide variety of growth rates, cell sizes, and cell cycle phase distributions. Population doubling times ranged from 1.3 h to 6.6 h, with cells growing fastest on YPD and slowest on non-fermentable minimal growth medium (SDGE) (Fig. 1A). The mean cell volume was largest in YPD at about 62 fL and smallest in SCGE at 46 fL (Fig. 1B). While overall, our results support the common notion that cells typically grow larger and faster in a nutrient-rich medium, the relationships between cell volume and doubling time across conditions are more complex and do not always follow this trend (Fig. EV1A). To determine the fraction of cells in different cell cycle phases, we quantified the DNA content with flow cytometry. The percentage of cells in the G1-phase notably increased in media with non-fermentable carbon sources, leading to smaller fractions of cells in S- and G2/M-phases (Fig. 1C).

To test if cells maintain a constant histone-to-DNA stoichiometry in the different nutrients, we measured the histone H2B protein levels by western blot analysis. For all media, we extracted total protein content from an equal number of cells and used Ponceau S staining for quantification (Fig. EV1B). Consistent with previous studies (Newman et al, 2006; Xia et al, 2022), we observe a decrease in total protein abundance for cells cultured in nutrient-poor media, consistent with their smaller cell size (Figs. 1D and EV1C). To compare the relative H2B protein expression in all growth media, we then normalized the histone protein amounts to total protein amounts. Across the different nutrients, we find that the histone H2B protein concentration decreases in inverse proportion to cell volume (Fig. 1E), suggesting a constant amount of H2B proteins per cell. This indicates that histone protein synthesis is coordinated with genomic DNA content rather than cell volume. In contrast, actin is maintained constant at a cell volume independent concentration (Fig. 1F). This is consistent with

the fact that actin expression scales with cell volume to ensure constant concentrations during growth (Claude et al, 2021; Swaffer et al, 2021).

### In different growth conditions, histone protein concentrations decrease with cell volume to maintain constant histone amounts

The data above show that histone protein concentrations decrease with cell volume to maintain constant amounts across nutrient conditions. To confirm that this is also true within a given cell cycle stage, we performed microfluidics-based live-cell microscopy and monitored the synthesis of histone H2B over time in individual haploid cells. For this purpose, we endogenously tagged *HTB1* and *HTB2*, the two genes encoding for the core histone H2B, with the fluorescent protein mCitrine. We then measured cell volume and total mCitrine intensity in newborn daughter cells by time-lapse microscopy across three nutrient conditions (Fig. 1G,H; Appendix Fig. S1).

In all growth media, H2B-mCitrine amounts per cell are constant during early G1 and increase approximately twofold during S-phase before they reach a plateau in G2M. As expected, the cell cycle duration strongly depends on the nutrient condition, with cells in SCGE exhibiting the longest cell cycle (Fig. 1H). Interestingly, the expression profiles indicate a similar timing of histone synthesis in all media, with a moderately increased synthesis duration in SCGE (Fig. EV1D,E). However, a (nutrient-dependent) maturation time of mCitrine may mask the underlying dynamics of histone synthesis. For all conditions, H2B protein concentration at birth decreases with cell volume because equal total histone amounts are produced in the mother cells (Fig. 1I,J). This was also confirmed by flow cytometry measurements of total H2B-mCitrine amounts in G1 in all seven growth media (Fig. EV1F). Thus, our experiments with fluorescently tagged H2B support the conclusion that histone protein amounts are maintained constant across nutrient conditions. However, we note that while the fluorescent tagging of both histone genes did not noticeably affect population doubling times, we detected increased mean cell volumes and elevated *HTB1* and *HTB2* transcript concentrations (Fig. EV1G–J). Thus, the exact mode of histone regulation in these strains has to be interpreted with care.

To examine how cells adjust the production of H2B-mCitrine immediately after a nutrient switch, we also performed a dynamic nutrient upshift experiment. To this end, we initially supplied cells growing in a microfluidic device with SCGE before switching the media to YPD. We imaged cells throughout the adaptation to the new environment and found that when exposed to dynamic nutrient shifts, cells still produce constant amounts of histones, suggesting tight regulation of histone homeostasis and rapid adaptation to the new nutrient environment (Fig. EV1K).

### Histone transcript concentrations are downregulated in poor nutrient conditions

Our results demonstrate that cells couple histone protein amounts to the genome content despite the nutrient-induced changes in cell growth and cell cycle progression. Since previous studies showed that the cell volume-independent coupling of histone amounts to DNA content is ensured at the transcript level (Claude et al, 2021; Swaffer

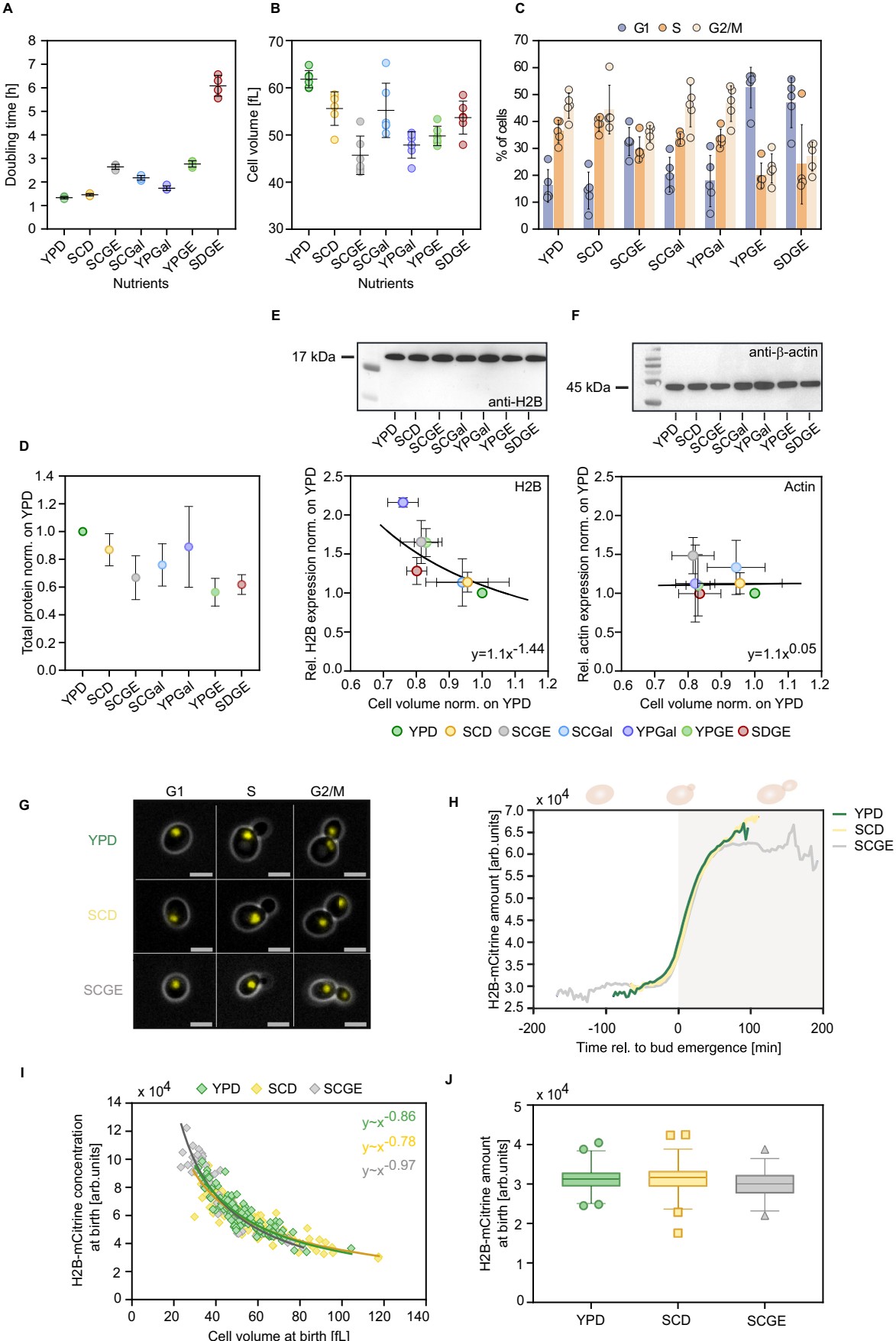

**Figure 1. Single-cell and population-level analyses reveal that histone protein concentrations decrease with cell volume across nutrient conditions, despite the nutrient-induced changes in cell growth.**

To determine the impact of different nutrients on the regulation of histone proteins, we grew cells on synthetic complete (SC), yeast peptone (YP), and minimal medium (SD), containing glucose (D) or galactose (Gal) as fermentable and glycerol and ethanol (GE) as non-fermentable carbon sources. (A) Doubling times are calculated from growth curves of exponentially growing cell populations in different growth media. Lines and error bars represent the mean and standard deviation of $n = 4$ independent replicates, each shown as an individual dot. (B) Mean cell volumes of cells growing in different nutrients, measured with a Coulter counter. Lines and error bars indicate mean values and standard deviations across $n = 6$ replicate measurements. (C) Flow cytometry analysis of the nutrient-dependent cell cycle distributions (percentage of cells in G1-, S-, and G2/M-phase) based on quantification of the DNA content. Means and standard deviations of $n = 5$ independent measurements are shown. (D) Total protein content, extracted from an equal number of cells in different growth media, was quantified by Ponceau S staining and normalized on YPD. Mean fold changes and standard deviations of at least 4 independent replicates are shown. (E, F) Protein bands of (E) histone H2B and (F) actin protein were quantified by western blot in different growth media and normalized to total proteins as determined from Ponceau stains. For each condition, total protein content was extracted from an equal number of cells. The relative protein expression is shown as a function of the relative nutrient-specific cell volume. The line shows fit with equation parameters obtained from linear regression on the double-logarithmic data. Mean and standard deviation of at least three biological replicates are shown. (G) Representative live-cell fluorescence and phase-contrast images of G1, S, and G2/M cells with *mCitrine*-tagged H2B (both Htb1 and Htb2 tagged), growing in three nutrient conditions. The scale bars represent 5 µm. (H) Mean amounts of H2B-mCitrine during the first cell cycle of newborn cells in YPD ($n_{YPD} = 87$), SCD ($n_{SCD} = 83$), and SCGE ($n_{SCGE} = 55$). Single-cell traces are aligned at bud emergence ($t = 0$). (I) H2B-mCitrine concentrations at birth are plotted as a function of cell volume for three nutrient conditions ($n_{YPD} = 106$, $n_{SCD} = 113$, $n_{SCGE} = 66$). Line show fits with equation parameters derived from a linear regression on the double-logarithmic data. (J) H2B-mCitrine amounts at birth in the different growth media. Box plots represent the median and 25th and 75th percentiles, whiskers indicate the 2.5th and 97.5th percentiles, and symbols show outliers ($n_{YPD} = 106$, $n_{SCD} = 113$, $n_{SCGE} = 66$). Source data are available online for this figure.

et al, 2021), we asked whether the nutrient-dependent histone homeostasis observed here is also achieved at the transcript level. In that case, we would predict that histone transcript amounts would be constant across nutrient conditions, resulting in an increased mRNA concentration for cells cultured in nutrient-poor conditions according to their relatively smaller cell size. To test this, we performed RT-qPCR and quantified histone transcript concentrations of asynchronous cell populations relative to total RNA in three nutrient conditions (Figs. 2A,B and EV2A–G). More specifically, each RT-qPCR measurement was normalized on the rRNA *RDN18*, which constitutes the majority of total RNA and is expressed at constant levels relative to total RNA across nutrients (Fig. EV2A). In contrast to our expectations, we found that histone transcript concentrations are significantly decreased in poor compared to rich growth media. At the same time, cells maintain constant mRNA concentrations of the control genes *ACT1* and *MDN1*.

To better understand the impact of nutrient-specific cell cycle progression on histone transcript levels, we additionally examined populations of synchronized cells. Specifically, we arrested cells carrying a β-estradiol-inducible *CDC20* allele (Ewald et al, 2016) in mitosis and released them synchronously into the cell cycle to study cell cycle-dependent histone mRNA expression by RT-qPCR (Figs. 2C and EV2H). After release (time = 0 min), transcript concentrations of histone and control genes were quantified at defined time points throughout the cell cycle. Our results show distinct expression peaks for histones *HTB1* and *HTB2* in both nutrient conditions, with notably reduced transcript concentrations in poor compared to rich growth medium (Fig. 2D,E). For *MDN1*, on the other hand, no significant change in mRNA expression is observed between nutrients (Fig. 2F). Importantly, cells on SCGE are smaller than cells on YPD also at the time of peak histone expression, ruling out the possibility that the lower peak expression of H2B observed in SCGE is a result of relatively increased cell volume (Fig. EV2I). Note that we surprisingly found that *ACT1* mRNA concentrations were decreased in poor growth medium (Fig. EV2J). However, we observed a similar decrease of *ACT1* mRNA when we measured the concentration in asynchronous cultures, suggesting that this is specific to the strain carrying the inducible *CDC20* allele and not due to cell cycle dependence (Fig. EV2K).

Our findings indicate that while the amount of histone proteins is tightly coupled to the DNA content, histone mRNA expression shows an unexpected dependence on the nutrient conditions. The apparent decoupling of histone protein and mRNA abundance suggests that medium-specific regulation of protein translation or stability is required to ultimately ensure histone homeostasis in different nutritional environments.

## Transcription accounts for the nutrient-dependent downregulation of histone transcripts in nutrient-poor conditions

As mRNA concentration is set by transcription and mRNA degradation, we sought to determine the contribution of these opposite reactions to the observed nutrient-dependent regulation of histone mRNA levels. More precisely, we studied histone and *ACT1* mRNA stability in rich and poor medium by inhibiting global transcription with thiolutin and monitoring the remaining mRNA by RT-qPCR over time. We then fitted a single exponential function to the mRNA decay curves to determine the mRNA half-lives (Fig. 2G). Consistent with previous studies (Bhagwat et al, 2021; Trcek et al, 2011), we found that histone mRNAs in YPD have short half-lives, whereas *ACT1* mRNA is more stable. Moreover, we showed that both histone and *ACT1* mRNAs are significantly more stable in poor compared to rich nutrient conditions. Again, this is consistent with previous studies that showed that transcription and degradation rates of many genes tend to increase in fast growth conditions, ensuring constant mRNA concentrations (García-Martínez et al, 2016a, 2016b).

Taken together, our results suggest that while the nutrient environment affects both histone mRNA synthesis and degradation, it is histone transcription that accounts for the nutrient-dependent downregulation of histone transcripts in nutrient-poor conditions.

## Histone promoter determines nutrient-dependence of histone transcript concentrations

Previously, we have shown that histone promoters can mediate the coordination of histone transcripts with genomic DNA content

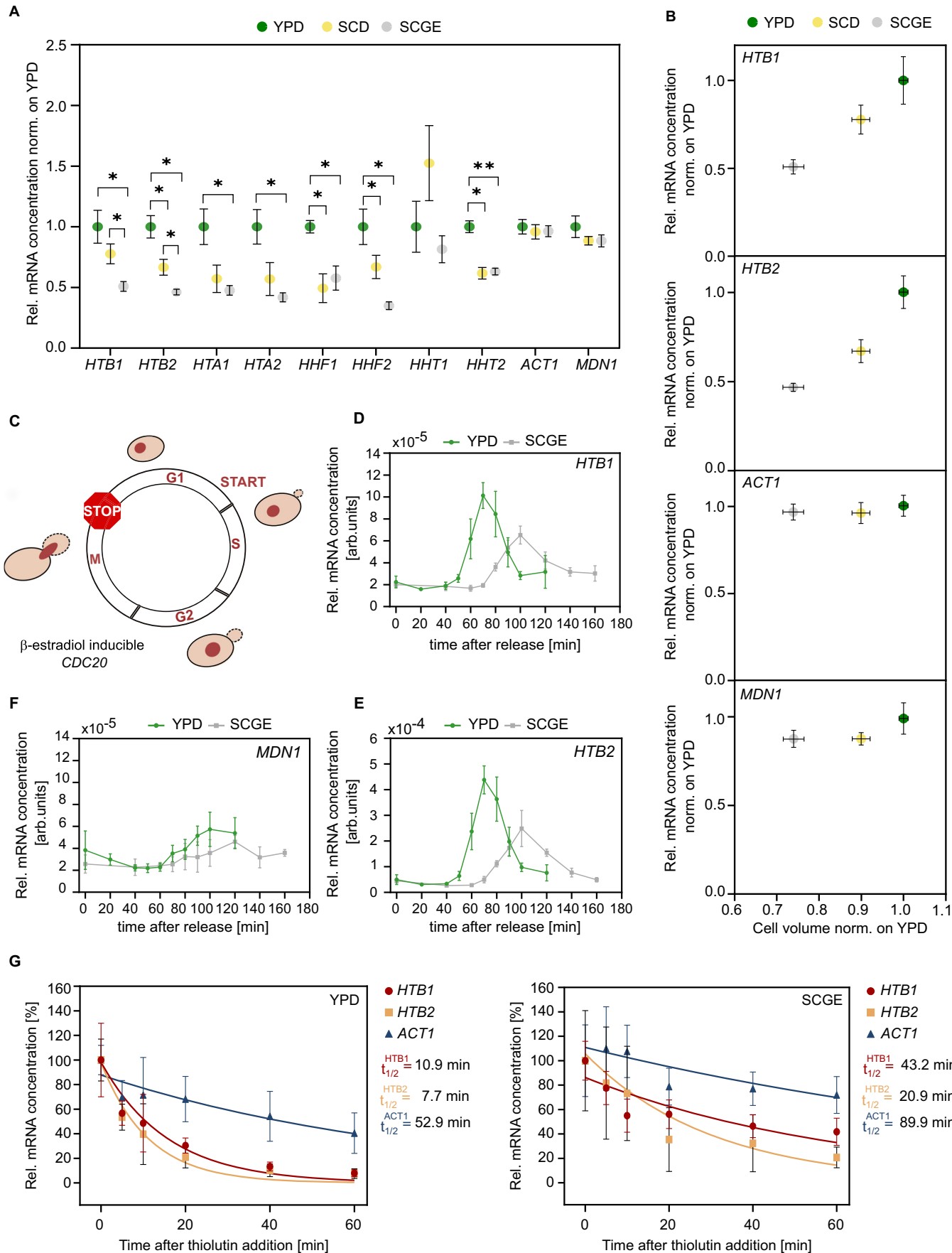

**Figure 2. Histone transcription is downregulated in poor nutrients.**

(A) RT-qPCR was used to quantify the mRNA concentrations of the core histone genes and the control genes *ACT1* and *MDN1* in different nutrient conditions. mRNA concentrations were normalized on *RDN18* and are shown as mean fold changes compared to YPD. Error bars indicate standard errors of at least four independent biological replicates. Significances were determined by an unpaired, two-tailed *t*-test for datasets that follow a Gaussian distribution or a Mann–Whitney test for datasets that are not normally distributed; *HTB1* (*$p_{YPD-SCGE}$ = 0.012, *$p_{SCD-SCGE}$ = 0.030); *HTB2* (*$p_{YPD-SCGE}$ = 0.012, *$p_{YPD-SCD}$ = 0.026, *$p_{SCD-SCGE}$ = 0.034); *HTA1* (*$p_{YPD-SCGE}$ = 0.030); *HTA2* (*$p_{YPD-SCGE}$ = 0.034); *HHF1* (*$p_{YPD-SCGE}$ = 0.015, *$p_{YPD-SCD}$ = 0.028); *HHF2* (*$p_{YPD-SCGE}$ = 0.014, *$p_{YPD-SCD}$ = 0.034); *HHT2* (**$p_{YPD-SCGE}$ = 0.0013, *$p_{YPD-SCD}$ = 0.028). (B) Relative mRNA concentrations of *HTB1*, *HTB2*, and *ACT1* as a function of the relative nutrient-specific cell volume. The mean and standard error of at least four biological replicates are shown. (C) Cells carrying *CDC20* under the control of a β-estradiol-inducible promoter were synchronized in mitosis. In the absence of β-estradiol, the expression of Cdc20 was turned off, preventing the cells from entering anaphase and exiting mitosis. (D–F) mRNA concentrations of *HTB1*, *HTB2*, and *MDN1* (normalized to *RDN18*) were measured by RT-qPCR after synchronous release into the cell cycle (*t* = 0) triggered by the addition of 200 nM β-estradiol. Mean and standard deviation of four biological replicates are shown. (G) Transcription inhibition experiments suggest that histone mRNA stability increases in poor nutrient conditions. mRNA half-lives of *HTB1*, *HTB2*, and *ACT1* were determined by adding the RNA polymerase inhibitor thiolutin to cells growing in different growth media and then measuring mRNA concentrations (normalized on *RDN18*) over time by RT-qPCR. Relative mRNA concentrations were normalized on the initial concentration at time = 0. For each time point, the mean and standard deviation of at least four biological replicates is plotted. Lines show single exponential fits to the individual data points of all replicates. Source data are available online for this figure.

despite changes in cell volume (Claude et al, 2021). We, therefore, asked whether histone promoters are also sufficient for the nutrient-dependent regulation of histone transcript concentrations. To test this, we created strains with an endogenously integrated *mCitrine* reporter driven by an additional *HTB1*, *HTB2*, or *ACT1* promoter (including the 5′ UTRs), respectively (Fig. 3A). Indeed, we found that the reporter mRNA concentrations decreased in nutrient-poor conditions, similar to the endogenous histone concentrations (Fig. 3B). However, our results reveal more elevated reporter mRNA concentrations in SCD, indicating that the promoter is not sufficient to capture the difference between YPD and SCD. Lastly, if expressed from the *ACT1* promoter, the *mCitrine* transcripts are kept at constant, nutrient-independent concentrations.

## Histone transcript amounts are independent of cell volume in each different nutrient condition

So far, we have established that histone promoters can be sufficient to mediate the nutrient-dependent regulation between fermentable and non-fermentable carbon source, but it is unclear whether the histone mRNA synthesis is uncoupled from cell volume for cells grown on each medium. To test this, we performed single-molecule fluorescence in situ hybridization (smFISH) combined with wide-field fluorescence microscopy, which allows studying cell volume- and cell cycle-dependent gene expression in different nutrient conditions on a single-cell level (Fig. 3A). We used bright-field images to determine cell volume and assigned each cell to a cell cycle stage based on the calculated bud-to-mother volume ratio. First, we quantified the nutrient-dependent mRNA concentrations of *ACT1* and *MDN1*, two representatives of scaling gene expression (Claude et al, 2021; Swaffer et al, 2021). As anticipated, both genes were continuously expressed throughout the cell cycle at constant concentrations (Figs. 3C and EV3A–D), and transcript amounts were increasing proportionally to cell volume (Fig. 3D). Moreover, we found similar mRNA copy numbers at a given cell volume regardless of the nutrient condition. This highlights the importance of maintaining appropriate concentrations despite the nutrient-induced changes in cell growth.

In contrast to *ACT1* and *MDN1*, histone biogenesis is regulated in a cell cycle-dependent manner, as it is tightly coupled to DNA replication. Consequently, histone genes are transcriptionally active in late G1 and S-phase (Hereford et al, 1981; Eriksson et al, 2012; Kurat et al, 2014b). Previous studies have shown that histone promoters are sufficient to mediate periodic transcription, resulting in an mRNA expression peak during the early-to-mid S-phase,

which is coordinated with genome content rather than cell volume (Claude et al, 2021; Osley et al, 1986; Eriksson et al, 2011). Using smFISH, we quantified the nutrient-dependent mRNA concentrations of *mCitrine* expressed from the *HTB1* or *HTB2* promoter (Figs. 3E,F and EV3E,F). In all growth conditions, we observe increased *mCitrine* mRNA concentrations in S-phase, before they eventually decline as cells progress further through the cell cycle (Fig. 3F). We also find that for each nutrient condition, the mRNA peak expression during S-phase is uncoupled from both cell volume and nuclear volume, leading to constant transcript amounts per cell. Yet, in nutrient-poor conditions, the mRNA copy number at a given cell or nuclear volume is significantly lower than in rich conditions (Figs. 3G and EV3G–I). The nutrient-dependent decrease of mRNA abundance in the S-phase in poor nutrients is consistent with our RT-qPCR-based results quantifying average mRNA concentration in asynchronous populations. Further confirming the distinct nutrient-dependence mediated by histone promoters, we found that if expressed from an *ACT1* promoter, *mCitrine* transcript amounts per cell increase with cell volume in all three conditions, but are independent of the nutrient condition— similar to the transcripts of endogenous *ACT1* (Fig. EV3J).

## Histone promoter truncation changes the nutrient-dependence of transcript concentrations

In *Saccharomyces cerevisiae*, the expression of core histones is controlled by positive and negative regulatory elements in the promoter regions. Positive transcriptional regulation is mediated by transcription factors, including Spt10 and SBF that bind to the upstream activating sequences (UASs) and activate transcription periodically. In contrast, the histone regulatory (HIR) complex acts as a negative transcriptional regulator that represses histone gene expression by binding to the NEG element, present in all core histone promoters except the *HTA2–HTB2* promoter pair (Kurat et al, 2014b; Eriksson et al, 2012).

Previously, we found that decreasing promoter strength can alter the cell volume-dependence of histone promoters, resulting in a promoter-mediated scaling of gene expression with cell volume (Claude et al, 2021). Specifically, a truncated *HTB1* promoter, consisting only of 300 bp and lacking part of the UASs and the NEG element, drives the expression of mCitrine in a cell volume-dependent manner. Motivated by these findings, we sought to determine whether different truncations of the *HTB1* promoter also induce changes in the nutrient-dependence of the *mCitrine* mRNA levels. For this purpose, we used haploid strains

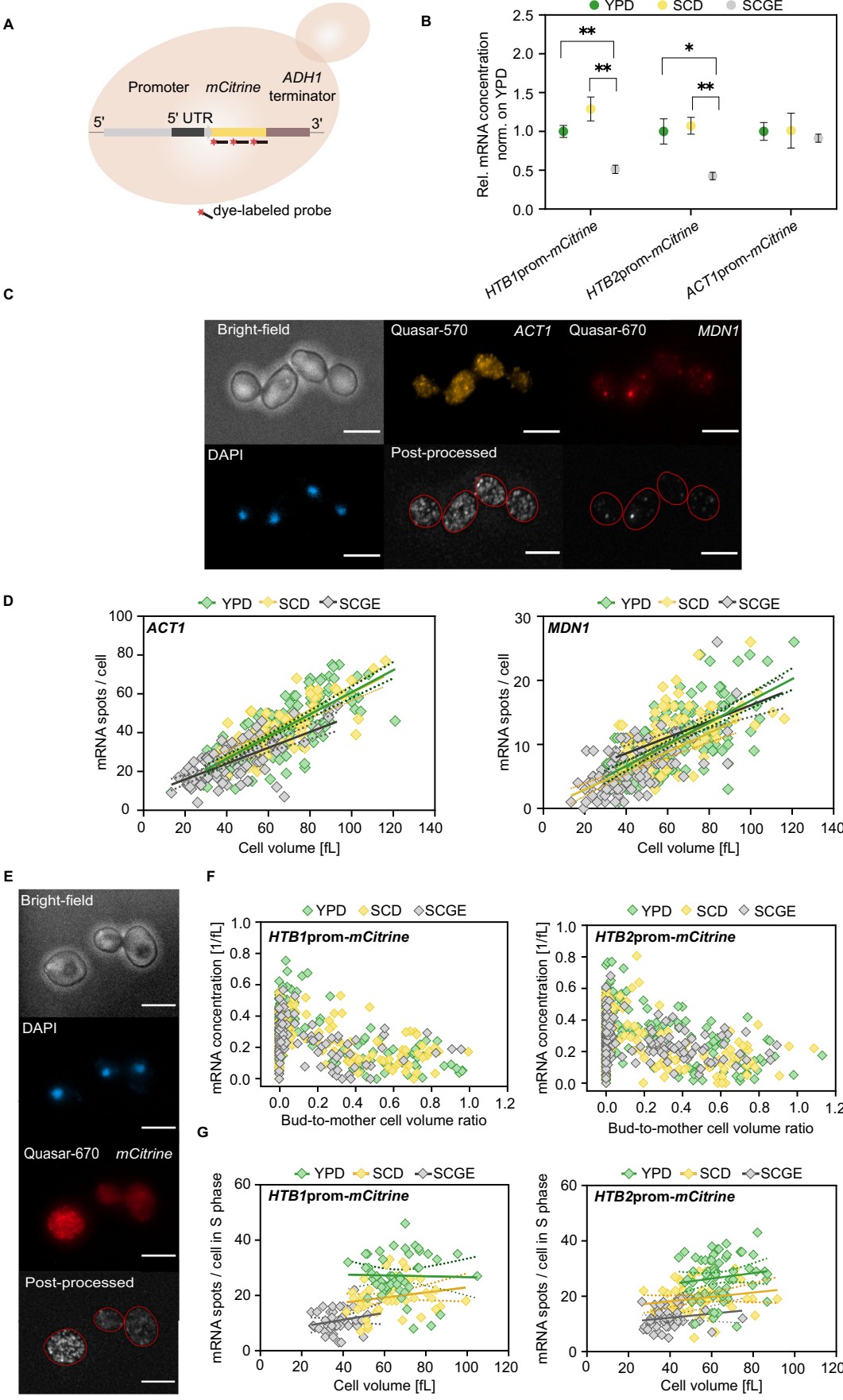

**Figure 3. Transcript amounts expressed from histone promoters are independent of cell volume in all nutrient conditions.**

(A) An *mCitrine* reporter gene driven by the promoter of interest (including the 5′ UTR) and regulated by the *ADH1* terminator was endogenously integrated into the *URA3* locus of wild-type haploid cells. smFISH was performed to study cell volume- and cell cycle-dependent gene expression in different nutrient conditions on a single-cell level (Fig. EV4). Schematic representation of fluorescently labeled probes binding to the target mRNA. (B) Relative mRNA concentrations of *mCitrine* expressed from the *HTB1*, *HTB2*, or *ACT1* promoter as determined by RT-qPCR. Mean fold changes with respect to YPD and standard errors of at least four replicate measurements are shown. Statistical significance was calculated by an unpaired, two-tailed *t*-test; *HTB1prom-mCitrine* (**$p_{YPD-SCGE}$ = 0.0016, **$p_{SCD-SCGE}$ = 0.0025); *HTB2prom-mCitrine* (*$p_{YPD-SCGE}$ = 0.029, **$p_{SCD-SCGE}$ = 0.0031). (C) Epifluorescence microscopy was performed to detect *ACT1* and *MDN1* transcripts targeted with Quasar-570- (yellow) and Quasar-670-labeled (red) probes, respectively. Representative images of cells grown on YPD are shown. Cell nuclei were stained with DAPI (blue), and bright-field images were used to estimate cell volume. For mRNA quantification, images were post-processed as described in Materials and Methods. The scale bars represent 5 μm. (D) Nutrient-dependent mRNA amounts of *ACT1* and *MDN1* per cell ($n_{YPD}$ = 176, $n_{SCD}$ = 87, $n_{SCGE}$ = 98) as a function of cell volume. Lines show linear fits; dashed lines indicate the 95% confidence intervals. (E) Representative images of cells expressing *HTB1prom-mCitrine* in YPD. Transcripts were detected using probes labeled with Quasar-670 (red). Nuclear DNA was stained with DAPI (blue). For mRNA quantification, images were post-processed as described in Methods and Materials. The scale bars represent 5 μm. (F) mRNA concentration of *mCitrine* expressed from an additional *HTB1* ($n_{YPD}$ = 144, $n_{SCD}$ = 158, $n_{SCGE}$ = 95) or *HTB2* promoter ($n_{YPD}$ = 161, $n_{SCD}$ = 194, $n_{SCGE}$ = 170) plotted against the corresponding bud-to-mother cell volume ratio in different nutrients. (G) Number of *mCitrine* mRNA spots per cell (*HTB1prom-mCitrine*: $n_{YPD}$ = 50, $n_{SCD}$ = 51, $n_{SCGE}$ = 39; *HTB2prom-mCitrine*: $n_{YPD}$ = 64, $n_{SCD}$ = 59, $n_{SCGE}$ = 50) as a function of cell volume during S-phase. Here, cells with a bud-to-mother volume ratio <0.3 were considered to be in S-phase. Lines show linear fits; dashed lines indicate the 95% confidence intervals. Source data are available online for this figure.

expressing mCitrine driven by a 450 and a 300 bp *HTB1* promoter (including the 5′ UTR), respectively, each truncated from the 5′-end (Claude et al, 2021) (Fig. 4A).

Our analysis revealed that similar to the full-length promoter, the 450 bp truncation showed reduced *mCitrine* mRNA concentrations in poor compared to rich conditions (Fig. 4B). In contrast, for the 300 bp truncation, we found that *mCitrine* mRNA concentrations considerably decreased in rich growth medium (Fig. 4C), becoming significantly lower than in poor medium (Fig. 4B). All strains tested in this experiment exhibited similar cell volumes and doubling times in the respective growth media (Appendix Fig. S2A,B). Our results, therefore, imply that the loss of the 150 bp sequence between the 450 bp and 300 bp truncation of the *HTB1* promoter leads to a marked change in the nutrient-dependence of the *mCitrine* transcripts. Given that this 150 bp sequence includes the UAS1 and UAS2 as well as the NEG element, it is possible that those regulatory elements contribute to the nutrient-dependent regulation of histone transcripts.

## Role of Spt10 in the nutrient-dependent regulation of transcript concentrations

We next asked whether the transcription factor Spt10, which specifically binds to the UASs, is required for the decrease of histone mRNA levels in poor growth media. To test this, we constructed a haploid strain carrying an additional copy of the *HTB1* promoter driving *mCitrine* expression, in which we mutated the Spt10 binding sites of the UAS3 and UAS4 elements (Eriksson et al, 2011) (Fig. 4A). We then quantified the *mCitrine* mRNA concentrations by RT-qPCR and found that mutation of the two UASs led to a weaker —albeit not completely abolished—nutrient-dependence (Fig. 4D). This is because the transcript levels measured for the mutant strain decreased significantly in YPD compared to the reference strain, but not in SCGE (Fig. 4E).

## Histone promoter is sufficient to compensate for nutrient-dependent histone transcript regulation

We have shown that despite the nutrient-dependence of histone transcript concentrations, cells maintain a constant histone-to-DNA stoichiometry across changing environments. This suggests that

additional regulation of translation efficiency or protein stability is required to achieve constant histone amounts across different nutrient conditions. We next asked whether histone promoters can also mediate this regulation at the protein level. We tested this by performing microfluidics-based live-cell microscopy and measuring the protein amounts of mCitrine expressed from the *HTB1* and *HTB2* promoter, respectively. In contrast to the endogenous histones, which are evenly distributed between the mother and daughter cell during division, mCitrine is partitioned along with the cytoplasm, in proportion to cell volume (Swaffer et al, 2021). Due to the asymmetric division of budding yeast, differently sized G1 cells could therefore inherit different amounts of mCitrine. Consequently, we refrained from comparing mCitrine amounts at birth and instead calculated the amounts produced during the cell cycle in rich and poor growth medium (Fig. 5A). Our results reveal that while the mRNA amounts of *mCitrine* are more than twofold lower in SCGE compared to YPD medium (mean fold change *HTB1prom-mCitrine* = 2.46, mean fold change *HTB2prom-mCitrine* = 2.18) (Figs. 3G and EV3G,H), the produced protein amounts are much more similar between conditions (mean fold change *HTB1prom-mCitrine* = 1.35, mean fold change *HTB2prom-mCitrine* = 1.13) (Fig. 5B,C). Thus, consistent with the experimental findings for the endogenous histones, the nutrient-dependence of mCitrine proteins expressed from H2B promoters is uncoupled from the mRNA levels. The fact that this regulation occurs also for the mCitrine reporter suggests that it is nutrient-dependent regulation of translation rather than regulated histone stability that compensates for the transcriptional downregulation in poor nutrients. To further confirm this, we analysed the stability of histone proteins in the rich and poor medium by constructing a strain carrying an extra copy of *HTB1-mCitrine*, expressed from a β-estradiol-inducible promoter (Fig. EV5A,B; Appendix Fig. S2C,D). Protein synthesis is initiated in hormone-supplemented growth medium, and can be turned off upon removal of β-estradiol, allowing the study of nutrient-dependent protein degradation over time. Our results reveal that Htb1 is a long-lived protein with reduced relative stability in poor compared to rich nutrients, when normalized to the doubling time in the respective media. Since histone transcripts are significantly less abundant in poor nutrients, these findings further support that regulated histone degradation does not ensure histone protein homeostasis across nutrients.

Given the importance of the 5′ untranslated region (UTR) in controlling translation efficiency and protein abundance, we sought to

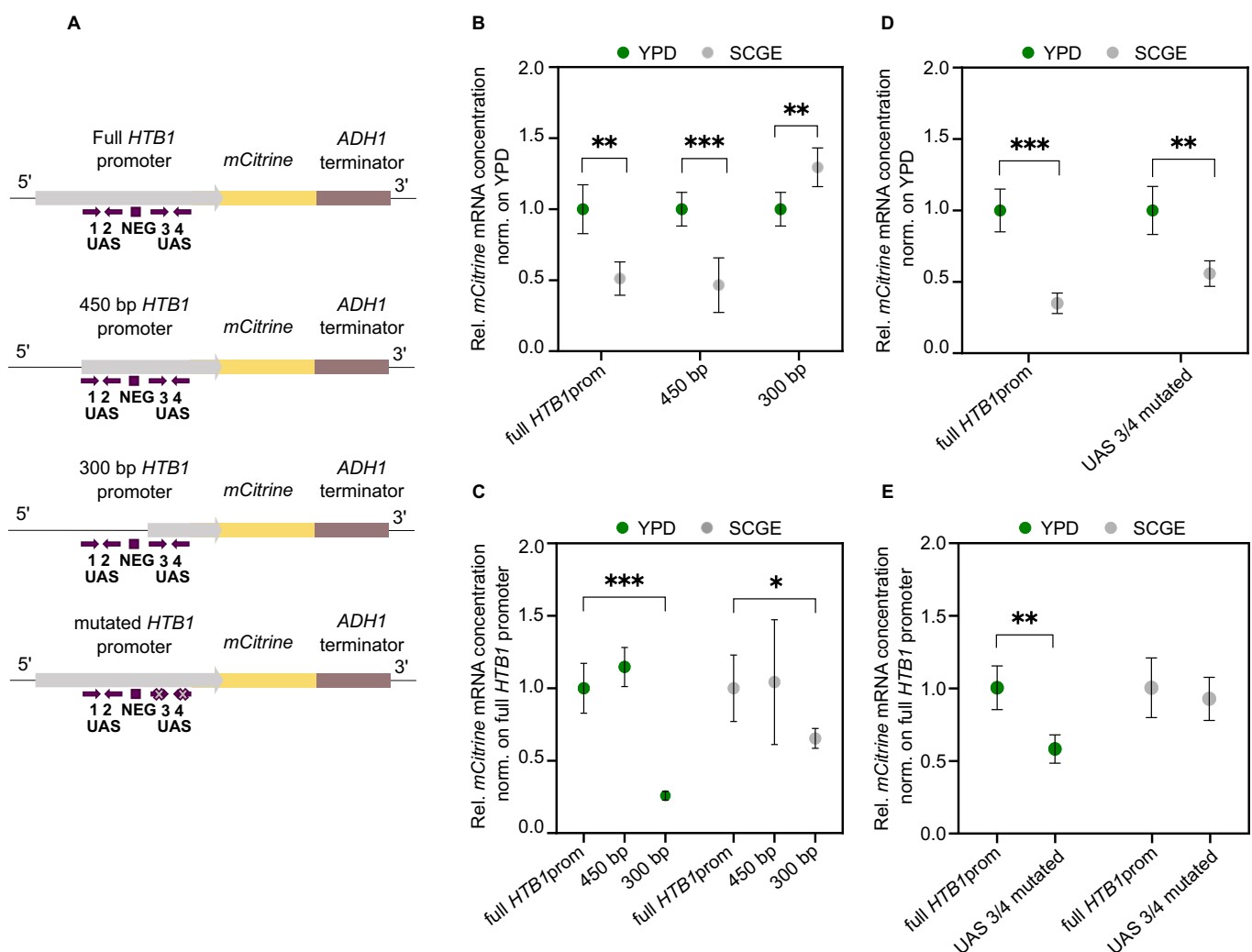

**Figure 4. Histone promoter truncation alters the nutrient-dependence of transcript concentrations.**

(A) Schematic representation of mCitrine expressed from the full *HTB1* promoter, the 450 bp and 300 bp truncations, as well as the *HTB1* promoter with mutated Spt10 binding sites in UAS3 and UAS4. Arrows indicate the location and orientation of the UAS elements, and the boxes show the NEG region. (B) RT-qPCR analysis of the *mCitrine* mRNA concentrations (normalized on *RDN18*) in different growth media. The results are shown as fold changes with respect to YPD in the respective media. Mean values and standard deviations across $n = 5–6$ independent replicates are shown; $**p_{full\ HTB1prom} = 0.0016$; $***p_{450bp} = 0.0008$; $**p_{300bp} = 0.0045$. (C) Quantification of *mCitrine* mRNA concentrations (normalized on *RDN18*) by RT-qPCR. The results are shown as mean fold changes compared to the reference strain carrying the full *HTB1prom-mCitrine* construct. Error bars indicate standard deviations across $n = 5–6$ independent replicates; ($***\ p_{fullHTB1prom−300bp}^{YPD} = 0.0008$); ($*p_{fullHTB1prom−300bp}^{SCGE} = 0.037$). (D, E) RT-qPCR analysis of the nutrient-dependent *mCitrine* mRNA concentrations (normalized on *RDN18*) measured for the UAS3/4 mutant. The results are shown as mean fold changes with respect to YPD (D) and to the reference strain (E) carrying the full *HTB1prom-mCitrine* construct. Error bars indicate standard deviations across $n = 4–5$ independent replicates. Different reference strains (full *HTB1* promoter) were used for the promoter truncation series (B, C) and the UAS3/4 mutant (D, E), respectively. Statistical significance was calculated by an unpaired, two-tailed $t$-test; Full *HTB1prom* ($***p_{YPD-SCGE} = 5.05 \times 10^{-5}$); UAS3/4 mutated ($**p_{YPD-SCGE} = 0.0041$); $**\ p_{fullHTB1prom−UAS3/4mutated}^{YPD} = 0.0015$. Source data are available online for this figure.

examine the influence of the *HTB1* 5′ UTR on the nutrient-dependent mCitrine expression from the *HTB1* promoter. To this end, we replaced the *HTB1* 5′ UTR with the 5′ UTRs of *MDN1* and *RPB4*, respectively (Fig. 5D; Appendix Fig. S2E,F; Appendix Table S1). *RPB4* is a subunit of the RNA polymerase II complex, and its expression has been reported to scale with cell size (Swaffer et al, 2023). If sequences within the *HTB1* 5′ UTR contribute to maintaining the histone protein amounts constant in rich and poor medium, then these replacements could lead to a change in the protein regulation. First, we performed RT-qPCR and found that replacing the 5′ UTR had no significant effect on the nutrient-

dependence of the *mCitrine* mRNA concentrations, which were decreased in nutrient-poor conditions (Fig. 5E). Moreover, while flow cytometry analysis of mCitrine protein expression revealed an overall decrease in the protein abundance after substitution of the *HTB1* 5′-UTR, the protein amounts remained constant between rich and poor conditions (Fig. 5F). This suggests that, despite its influence on protein abundance, the histone 5′ UTR is not required for nutrient-dependent histone homeostasis.

To gain further insight into the regulatory mechanisms underlying the decoupling of histone mRNA and protein abundance

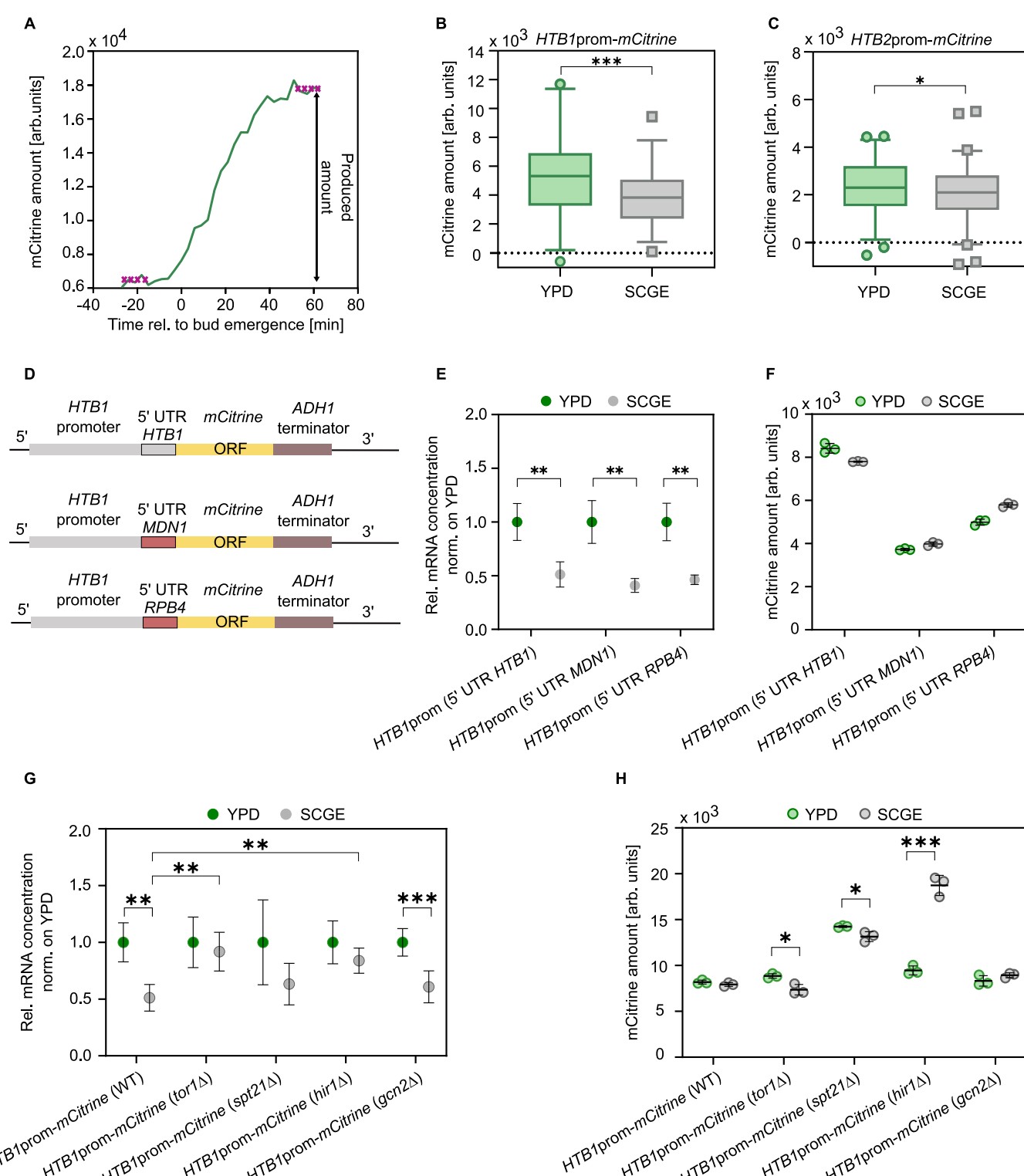

across nutrients, we investigated the effect of four carefully selected factors, Spt21, Hir1, Tor1, and Gcn2, on promoter-mediated histone expression in rich and poor growth medium. Spt21 serves as an important activator of histone gene regulation by interacting with the histone promoter (Bhagwat et al, 2021; Eriksson et al, 2012). On the other hand, Hir1 is part of the HIR complex that is

thought to repress histone gene expression through localization to the NEG region (Eriksson et al, 2012; Kurat et al, 2014a). Gcn2 is a protein kinase that phosphorylates the alpha-subunit of the eukaryotic translation initiation factor 2 (elF2) and coordinates protein synthesis in response to nutrient availability (Kubota et al, 2001; Murguía and Serrano, 2012). Similarly, the protein kinase

**Figure 5. mCitrine protein expression driven by H2B promoters is uncoupled from nutrient-dependent mRNA levels.**

(A) Protein amounts of mCitrine expressed from the *HTB1* and *HTB2* promoter during the cell cycle were measured in rich and poor growth medium using live-cell fluorescence microscopy. Representative fluorescence intensity trace of a haploid cell expressing *HTB1prom-mCitrine* in YPD. The mCitrine amounts produced during the cell cycle were calculated as the difference between the median of the first four time points and the median of the last four time points. (B, C) Protein amounts of mCitrine expressed from the *HTB1* and *HTB2* promoter were measured using live-cell fluorescence microscopy. Box and whisker plots represent the distribution of total mCitrine amounts produced during the cell cycle, under different nutrient conditions. Box plots represent median and 25th and 75th percentiles, whiskers indicate the 2.5th and 97.5th percentiles and symbols show outliers (*HTB1prom-mCitrine*: $n_{YPD} = 70$, $n_{SCGE} = 76$; *HTB2prom-mCitrine*: $n_{YPD} = 106$, $n_{SCGE} = 157$); *HTB1prom-mCitrine* (***$p_{YPD-SCGE} = 0.0004$); *HTB2prom-mCitrine* (*$p_{YPD-SCGE} = 0.039$). (D) To examine the influence of the *HTB1* 5′ UTR on the nutrient-dependent mCitrine expression from the *HTB1* promoter, the 5′ UTR was replaced with the 5′ UTRs of *MDN1* and *RPB4*, respectively. (E) RT-qPCR analysis of the *mCitrine* mRNA concentrations (normalized on *RDN18*) in different growth media. Results are shown as mean fold changes compared to YPD. Error bars indicate standard deviations of at least four independent replicates; *HTB1prom* (5′ UTR *HTB1*) (**$p_{YPD-SCGE} = 0.0016$); *HTB1prom* (5′ UTR *MDN1*) (**$p_{YPD-SCGE} = 0.0028$); *HTB1prom* (5′ UTR *RPB4*) (**$p_{YPD-SCGE} = 0.0022$). (F) mCitrine protein amounts as determined by flow cytometry. Lines and error bars represent mean values and standard deviations of three biological replicates. (G) The mRNA concentrations of *mCitrine* (normalized on *RDN18*) expressed from the *HTB1* promoter in wild-type, *tor1Δ*, *spt21Δ*, *hir1Δ*, and *gcn2Δ* cells were quantified by RT-qPCR in rich and poor medium. Mean fold changes with respect to YPD and standard deviations across $n = 5$–7 independent replicates are shown. The statistical significance of the fold change in transcription between YPD and SCGE for the *tor1Δ*, *spt21Δ*, *hir1Δ*, and *gcn2Δ* strains compared to the respective wild-type reference strain was tested using an unpaired, two-tailed *t*-test; *HTB1prom-mCitrine* (WT) (**$p_{YPD-SCGE} = 0.0016$, **$p_{tor1Δ} = 0.0028$, **$p_{hir1Δ} = 0.0037$); *HTB1prom-mCitrine* (gcn2Δ) (***$p_{YPD-SCGE} = 0.0002$). (H) Flow cytometry analysis of the mCitrine protein amounts expressed from the *HTB1* promoter in the different strains grown on rich and poor medium. Lines with error bars indicate mean values and standard deviation across $n = 3$ measurements; *HTB1prom-mCitrine* (tor1Δ) (*$p_{YPD-SCGE} = 0.014$); *HTB1prom-mCitrine* (spt21Δ) (*$p_{YPD-SCGE} = 0.026$); *HTB1prom-mCitrine* (hir1Δ) (***$p_{YPD-SCGE} = 0.0002$). Source data are available online for this figure.

Tor1 plays a crucial role in the coordination of cellular growth with nutrient availability, adjusting protein biosynthetic capacity according to the nutritional environment (Cardenas et al, 1999; Powers and Walter, 1999). To assess how cells lacking one of these four factors regulate the nutrient-dependent expression of *HTB1prom-mCitrine* transcripts and proteins, we performed RT-qPCR and flow cytometry analysis, respectively (Fig. 5G,H; Appendix Fig. S2G,H). Our RT-qPCR results showed that while the nutrient-dependent regulation of transcription is still intact in *spt21Δ* and *gcn2Δ* cells, deletion of *TOR1* or *HIR1* disrupted the nutrient-dependence of the reporter transcripts, leading to close to constant concentrations in rich and poor growth medium (Fig. 5G). Of note, as described above, we also observed a significant change in the nutrient-dependence of the *HTB1prom-mCitrine* transcripts for the 300 bp truncated *HTB1* promoter, which among others, lacked the NEG element required for HIR repression (Fig. 4B).

We then used flow cytometry to quantify the amounts of mCitrine protein expressed from the *HTB1* promoter and observed an upregulation of protein abundance in YPD in cells lacking *TOR1* (Fig. 5H). Thus, the lack of *TOR1* exerts a differential effect on the transcriptional and translational regulation of histones, leading to significantly altered transcript and protein levels across conditions. On the other hand, we found that deletion of *HIR1* resulted in significantly increased mCitrine amounts in SCGE, in line with the increased transcript amounts. This suggests that the absence of HIR-dependent repression has a stronger effect on histone transcript regulation in poor medium, ultimately disrupting histone protein homeostasis. Overall, these findings demonstrate the importance of Tor1 and Hir1 in achieving the correct regulation of histone transcripts and maintaining constant amounts of histone proteins across nutrients. Our flow cytometry analysis further revealed an overall increase in mCitrine protein abundance for *spt21Δ* cells grown on rich and poor growth medium compared to wild-type cells. Moreover, between conditions, mCitrine is more abundant in YPD. Interestingly, Spt21 has previously been reported to have a differential effect on the expression levels of *HTB1* and *HTB2* (Eriksson et al, 2012; Kurat et al, 2014a; Dollard et al, 1994; Hess and Winston, 2005; Chang and Winston, 2013). While *HTB2* mRNA levels were shown to drastically decrease upon deletion of

*SPT21*, *HTB1* transcript levels were unaffected or increased, consistent with our results. Lastly, cells lacking *GCN2*, were not affected in their ability to achieve histone protein homeostasis in rich and poor medium.

Overall, our findings indicate that while Spt21 affects histone protein abundance in rich and poor conditions, Tor1 and Hir1 play a significant role in the regulation of histone mRNA and protein expression across nutrients.

## Cells are more sensitive to excess histone accumulation under non-fermentable growth conditions

So far, we have characterized nutrient-dependent histone homeostasis at protein and transcript levels. We showed that cells maintain constant stoichiometry between DNA and histone proteins. Surprisingly, histone transcripts are less abundant in poor nutrient conditions, raising the question of why it could be beneficial for cells to downregulate histone transcripts even though protein amounts are maintained constant. Interestingly, a recent study showed that cells exhibit a lower tolerance to excess histones under glucose-limited conditions, resulting in decreased cell fitness (Bruhn et al, 2020). In our case, this could imply that cells in poor, i.e., non-fermentable, growth medium keep the histone transcript concentrations low to reduce the risk of accumulating excess histones. To assess the importance of nutrient-dependent transcript regulation, we tested how cells respond to aberrantly high levels of histone transcripts under fermentable and non-fermentable growth conditions. For this purpose, we followed the strategy of Bruhn et al, (Bruhn et al, 2020), and used a *rad53Δ* mutant, which is defective in histone transcript regulation and in the degradation of excess histones (Fig. 6A). Cells lacking Rad53, a DNA damage response kinase, are only rendered viable by additional deletion of the ribonucleotide reductase inhibitor Sml1. The *sml1Δ* mutant has a similar doubling time and cell volume compared to the wildtype and served as a reference (Appendix Fig. S2I,J). Analysis of the histone transcript concentrations in the *sml1Δrad53Δ* mutant revealed increased accumulation of *HTB2* and *HHT2* transcripts compared to the *sml1Δ* strain, on both rich and poor carbon sources (Fig. 6B,C). We included *HHT2* in our analysis as a control,

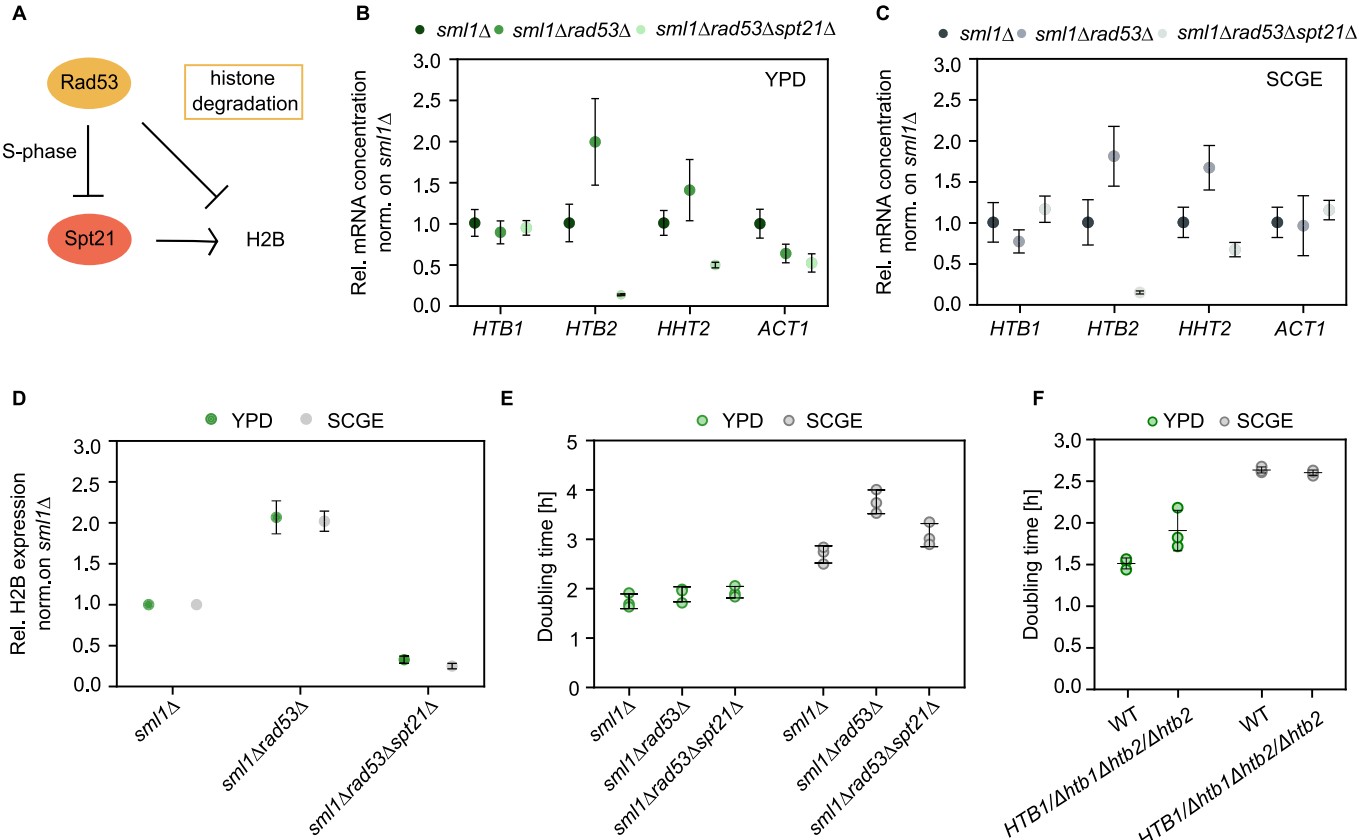

**Figure 6. Cells growing on non-fermentable carbon sources are more sensitive to histone overexpression than in nutrient-rich conditions.**

(A) Rad53 is required for the degradation of excess histones and also regulates histone levels by inhibiting the transcription activator Spt21. (B, C) mRNA concentrations of *HTB1*, *HTB2*, *ACT1*, and *HHT2* (normalized on *RDN18*) for *sml1Δrad53Δ* and *sml1Δrad53Δspt21Δ* cells are shown as mean fold changes with respect to *sml1Δ*, in YPD (B) and SCGE (C). Error bars indicate standard errors across n = 3–5 biological replicates. (D) H2B protein levels, normalized on total protein, as determined by western blot analysis. Mean fold changes with respect to *sml1Δ* and standard errors of at least four replicates are shown. (E) Doubling times calculated from growth curves of *sml1Δ*, *sml1Δrad53Δ*, and *sml1Δrad53Δspt21Δ* cells growing on fermentable and non-fermentable carbon sources. Lines with error bars represent the mean and standard deviation of n = 3 independent measurements. (F) Reduced histone amounts slow down cell growth in rich nutrients. Doubling times were calculated from growth curves of *HTB1/Δhtb1Δhtb2/Δhtb2* and wild-type diploid cells growing on fermentable and non-fermentable carbon sources. Lines and error bars represent the means and standard deviations of n = 3 independent measurements shown as individual dots. Source data are available online for this figure.

as it has been already shown by Bruhn et al (Bruhn et al, 2020) to exhibit elevated mRNA expression concentrations in *sml1Δrad53Δ* cells. However, *HTB1* transcript concentrations were not affected by deletion of *RAD53*. Despite that, western blot analysis revealed that *sml1Δrad53Δ* mutants have increased concentrations of H2B protein in both growth media (Fig. 6D). To examine whether this overexpression affects cell growth on rich and poor carbon source, we measured the corresponding doubling times (Fig. 6E). In SCGE, we observe that *sml1Δrad53Δ* cells grow significantly slower than the reference *sml1Δ* strain. In YPD, however, the lack of Rad53 does not cause major changes in the doubling time. In addition to accumulating excess histones, *sml1Δrad53Δ* mutants are also defective in DNA replication (Desany et al, 1998; Gunjan and Verreault, 2003; Tercero and Diffley, 2001) and DNA damage response (Zhao et al, 1998; Zhao, 2001; Gunjan and Verreault, 2003), among others. To disentangle the contribution of histone overexpression to the observed growth phenotype, we drastically decreased histone protein levels by deleting *SPT21*, a histone transcription activator, and examined the effects on cell growth. We

find that in SCGE, *sml1Δrad53Δspt21Δ* triple deletion mutants grow faster, indicating that the growth phenotype of *sml1Δrad53Δ* cells is partly rescued. In contrast, the doubling time in YPD is not noticeably affected (Fig. 6E). Thus, our results suggest that not only upon glucose limitation (Bruhn et al, 2020) but also during growth on non-fermentable carbon sources, cells exhibit higher sensitivity to excess histone accumulation than in rich glucose conditions. This suggests that on poor nutrients, histone transcription is downregulated to avoid toxic overexpression, but in rich nutrients, rapid histone production becomes more important. To further test this hypothesis, we created a diploid strain in which we deleted the endogenous alleles of *HTB2* and one allele of *HTB1* to examine the effects of decreased histone amounts on cell growth in rich and poor conditions. While cells in SCGE are unaffected in their doubling time, in YPD, we find that reducing the histone amounts leads to slower cell growth (Fig. 6F).

Taken together, our results suggest that nutrient-dependent regulation of histone transcripts is important because—depending on the nutrient condition—histone overexpression affects cells to

different degrees, which shifts the optimal balance between production speed and prevention of excess production.

## Discussion

Despite changes in cell growth and cell cycle progression, cells maintain constant histone-to-DNA stoichiometry across different nutrient conditions. This implies that histone protein concentrations are higher in poor compared to rich nutrients, to account for the smaller cell size. Paradoxically, histone transcripts show the opposite trend, and are downregulated in poor nutrients. The apparent decoupling of histone mRNA and protein nutrient-dependence suggests that histone-specific nutrient-dependent regulation of translation is required to ensure constant histone amounts.

Previous work on cell volume-dependent histone homeostasis showed that the coordination of histones with genome content is already established at the transcript level and is at least in part mediated by the promoter (Claude et al, 2021; Swaffer et al, 2021). This differential regulation of histones was proposed to be achieved through template-limited transcription, where the gene itself, rather than the polymerase (Swaffer et al, 2023), limits histone mRNA synthesis (Claude et al, 2021). Our single-cell analysis revealed that for each nutrient condition, transcript amounts expressed from a histone promoter are indeed independent of cell volume. However, the fact that transcript amounts decrease significantly in poor compared to rich growth media, suggests that nutrient-dependent histone regulation cannot be explained by 'gene-limited transcription' alone. Highlighting this distinct regulation of histones, we found that in contrast to histone mRNAs, the mRNA amounts of *ACT1* and *MDN1*, two representatives of scaling gene expression, increase in proportion to cell volume, but are largely independent of the nutrient condition.

As histone synthesis is restricted to the S-phase, it could, in principle, be possible that nutrient-induced changes in the relative duration of the cell cycle phases explain the decrease of histone transcripts in poor growth media. Cells growing on SCGE spend more time in G1 compared to growth on YPD, resulting in relatively shorter S- and G2/M-phases. However, quantification of histone mRNA in synchronized cell populations still showed downregulated mRNA levels of *HTB1* and *HTB2* in poor compared to rich medium at the time of maximal expression. This strongly suggests that the decreased mRNA concentration observed in asynchronous cultures is not simply a consequence of nutrient-dependent changes in the relative duration of the cell cycle phases. Ultimately, our findings suggest that nutrient-dependent transcription must account for the reduced transcript levels, as mRNA stability increases in poor conditions. Furthermore, we have shown that histone transcript regulation across the nutrients requires Hir1 regulation, which is likely mediated through the NEG element.

Why do cells downregulate histone transcripts if the amount of protein needs to be coordinated with the DNA content and, therefore, is kept constant across nutrient conditions? Previously, it has been shown that upon glucose limitation, cells are more sensitive to histone overexpression. This sensitivity is partly induced by the hyper-acetylation of excess histones, which, under poor growth conditions, affects the Ac-CoA-dependent metabolism to a greater extent, due to the lower availability of Ac-CoA (Bruhn

et al, 2020). We now show that also for cells growing on non-fermentable carbon sources histone overexpression is more toxic than for cells grown on glucose. Our findings, therefore, suggest that cells growing on poor nutrients reduce the risk of accumulating excess histones by maintaining low concentrations of histone transcripts. In rich nutrient environments, on the other hand, where cells exhibit higher growth rates, fast histone production may be more critical.

The protein-to-mRNA ratio is dictated by translation and protein degradation. Thus, the uncoupling of the regulation of protein and mRNA abundances we observed in different nutrients implies the need for nutrient-specific regulation of translation or protein stability. Specifically, our data suggest that to compensate for the decreased transcript concentrations in poor nutrients, the relative translation efficiency of core histones is higher than in rich nutrients. Moreover, we found that intact TOR-signaling is required to accurately regulate histone biogenesis in different nutrient environments.

We have shown that histone promoters mediate nutrient-dependent transcription as well as its compensation on the protein level even when expressing the fluorescent reporter mCitrine. This indicates that the decoupling of histone transcript and protein nutrient-regulation is achieved mainly through nutrient-dependent translation rather than protein degradation. Interestingly, our experiments revealed that the histone 5′ UTR is not required for maintaining constant amounts of proteins expressed from a histone promoter across changing nutrients. This indicates that histone translation is regulated through an "imprinting" mechanism: Previous studies have identified several factors that bind to newly produced mRNAs in the nucleus and remain associated with them throughout their lifecycle, controlling mRNA export, localization, decay or translation (Dahan and Choder, 2013). For example, it was proposed that Pol II remotely modulates mRNA translation and decay through co-transcriptional binding of the Rpb4/7 heterodimer to Pol II transcripts (Goler-Baron et al, 2008; Harel-Sharvit et al, 2010; Richard et al, 2021). Thereby in addition to the encoded information, mRNAs would carry information "imprinted" by Rpb4/7, which is required for proper post-transcriptional regulation. Moreover, several studies of promoter-dependent mRNA stability suggest that promoter elements can also contribute to mRNA imprinting, providing cross-talk with cytoplasmic processes (Trcek et al, 2011; Bregman et al, 2011).

Overall, we have characterized the distinct regulation of histone homeostasis in changing environments, highlighting the importance of cell cycle-dependent genes to maintain accurate protein concentrations despite the nutrient-induced changes in cell growth and cell cycle progression. Our work revealed a surprising mode of regulation, where histone protein concentrations are decoupled from transcript concentrations. We speculate that this allows cells to balance the need for rapid production under fast growth conditions with the tight regulation required to avoid toxic overexpression in poor media (Fig. 7A–C). More generally, this suggests that cells use separate regulation of transcripts and translation as a way to not only control the final protein concentration but also optimize the balance between production speed and accuracy. Future studies will be needed to reveal whether such regulation also occurs for other genes, in particular cell cycle-dependent genes that are high expressed only during a short fraction of the cell cycle.

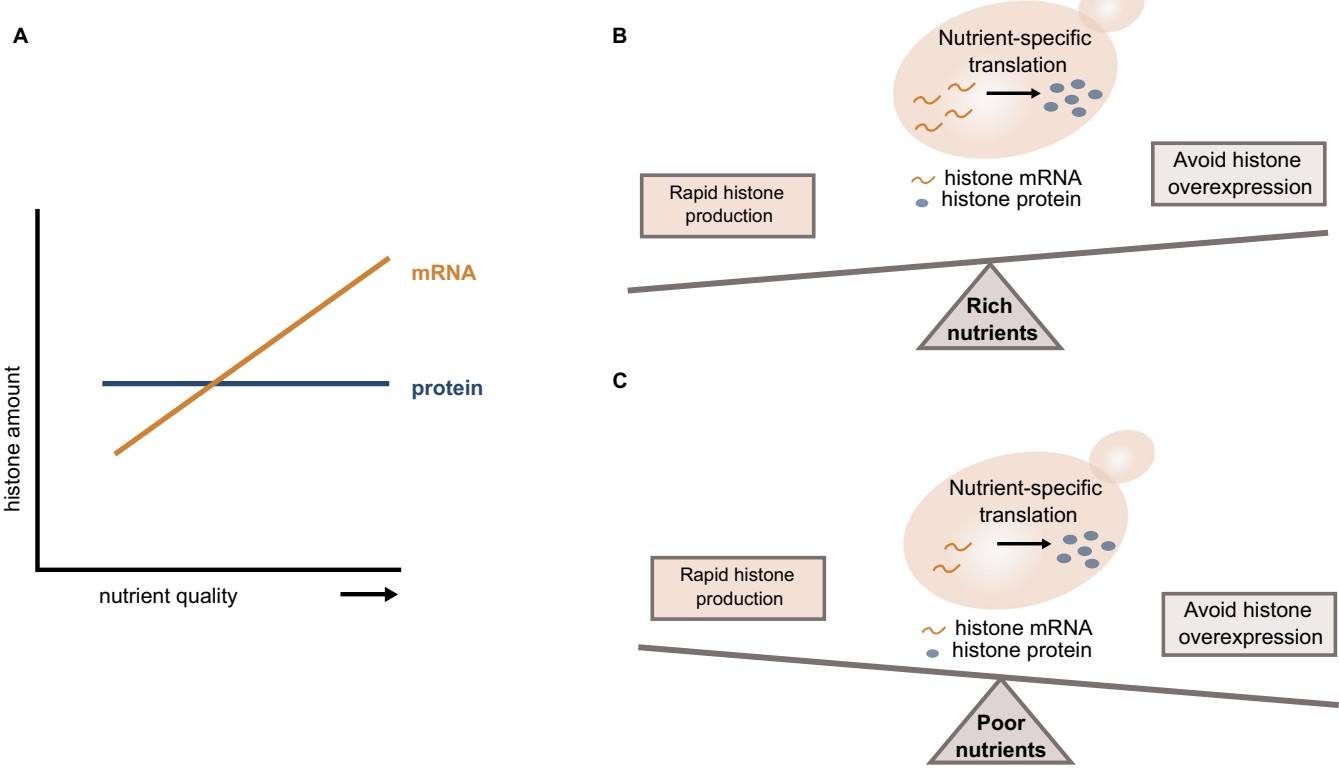

**Figure 7. Histone homeostasis in different nutrient environments.**

(A) While cells maintain constant amounts of histone proteins across changing environments, the histone mRNA expression is increased in nutrient-rich conditions. (B, C) Decoupling of protein and mRNA abundance is achieved through nutrient-dependent translation, allowing cells to balance the need for rapid histone production in rich growth media (B) with the tight regulation required to prevent histone overexpression in poor growth media (C).

## Methods

### Strains and culture conditions

Budding yeast strains used in this study are haploid derivatives of W303 and were constructed using standard procedures. All transformants were validated using PCR and sequencing. Full genotypes for each strain can be found in Appendix Table S2, and strains are available upon reasonable request.

Cells were grown under different nutrient conditions at 30 °C in a shaking incubator at 250 rpm (Infors, Ecotron). Yeast colonies were inoculated in 4 mL yeast peptone medium containing 2% glucose (YPD) and were cultivated for at least 6 h at 30 °C before being washed and transferred to synthetic complete (SC), yeast peptone (YP), or minimal medium (SD), with either glucose (D) or galactose (Gal) as fermentable or glycerol and ethanol (GE) as non-fermentable carbon sources. Cells were then grown in the respective growth medium for at least 18 h to $OD_{600} = 0.3$–0.9. Through appropriate dilutions, cell density was maintained below $OD_{600} = 1$. Optical densities were measured using a spectrophotometer (NanoDrop One$^C$, Thermo Fisher Scientific).

For cell cycle synchronization, cells carrying *CDC20* under the control of a β-estradiol-inducible promoter were arrested in mitosis. To this end, pre-cultures were grown on YPD supplemented with 80 nM β-estradiol. Cells were then transferred to YPD or SCGE containing 80 nM β-estradiol and cultured for at least 18 h to $OD_{600} = 0.3$–0.9. After washing twice with 3 mL nuclease-free water, cells were arrested in hormone-free YPD or SCGE for 2 or 3 h, respectively. Following the addition of 200 nM β-estradiol, cells were synchronously released into the cell cycle.

### Western blot

Total protein extracts were prepared according to a previously established protocol (Kushnirov, 2000). Briefly, cell cultures (25 mL) were grown in different growth media for at least 18 h to ensure steady-state conditions (see above). Prior to harvesting, cell volume distributions and cell numbers per mL were determined using a Coulter Counter (Beckman Coulter, Z2 Particle Counter). From each culture, $5 \times 10^7$ cells were collected by centrifugation (4k rpm, 3 min) and washed with 1 mL of ice-cold double-distilled water before being spun down (10k rpm, 2 min) and resuspended in 400 µL of 0.1 M NaOH. After incubation at room temperature (RT) for 10 min, cells were again pelleted (10k rpm, 2 min), and then boiled (3 min, at 95 °C) in 120 µL of reducing 1x LDS sample buffer, containing 30 µL of 4X Bolt™ LDS sample buffer (Invitrogen), 12 µL of 10X Bolt™ sample reducing agent (Invitrogen) and 78 µL of double-distilled water.

Following centrifugation (10k rpm, 2 min), 5–10 µL of total protein extracts (supernatant) were loaded into each lane of

commercially available Bolt™ 12% Bis-Tris plus mini-gels (Invitrogen). Gels were run (200 V, 160 mA, 20–25 min) in 1X Bolt™ MES SDS Running Buffer (Invitrogen), and the separated proteins were then transferred onto nitrocellulose membranes (10 V, 160 mA, 60 min) using the mini-blot-module (Invitrogen). In the next step, membranes were stained with Ponceau S, and total proteins were visualized using the ChemiDoc™ MP imaging system (Bio-Rad). To detect proteins of interest, membranes were blocked in TBST (Tris-buffered saline, 0.2% Tween 20) with 5% milk and incubated overnight at 4 °C with the primary antibodies: rabbit monoclonal anti-histone H2B (Abcam Cat#ab188291, 1:2000) or mouse monoclonal anti-beta actin (Abcam, Cat# ab170325, 1:10,000). After washing the membranes in TBST (3 ×5 min), they were probed (1.5 h, RT) with the HRP-conjugated secondary antibodies: goat anti-mouse IgG (Abcam Cat# ab205719, 1:10000) or goat anti-rabbit IgG (Abcam Cat# ab205718, 1:10000). To visualize the protein bands, membranes were incubated (5 min, RT) in Clarity™ western ECL substrate (Bio-Rad) and imaged using the ChemiDoc™ MP imaging system (Bio-Rad). Quantification of band intensities was carried out using the Image Lab 5.2.1 software (Bio-Rad).

## RNA extraction and RT-qPCR

Total RNA was extracted from $2–5 \times 10^7$ cells grown in different nutrients (see above) with the YeaStar RNA Kit (Zymo Research) following the manufacturer's instructions. Prior to harvesting, cell numbers per mL were measured using a Coulter Counter. Concentration and purity of the eluted RNA were determined with a spectrophotometer (NanoDrop One$^C$, Thermo Fisher Scientific) before 800 ng total RNA was reverse transcribed using random primers and the high-capacity cDNA reverse transcription kit (Thermo Fisher Scientific). To measure relative mRNA levels of target genes, the obtained cDNA was diluted 10-fold (*HTB2, ACT1, MDN1, mCitrine*) or 100-fold (*HTB1, HTA1, HTA2, HHT1, HHT2, HHF1, HHF2*) in double-distilled water and 2 µL of the dilutions were used as templates for quantitative PCR (qPCR). All qPCR reactions were performed on a LightCycler 480 Multiwell Plate 96 (Roche) using the SsoAdvanced Universal SYBR Green Supermix (Bio-Rad) and target-specific primers (Appendix Table S3). For each target gene, mean $Cq^{Gene}$ values of three technical replicates per sample were normalized to the reference gene *RDN18*, and relative mRNA concentrations were calculated by the formula: $\log_2$ (relative concentration) = $-(Cq^{Gene} – Cq^{RDN18})$.

## mRNA stability measurements

Cell cultures (50 mL) were grown in YPD and SCGE, respectively, for at least 18 h to $OD_{600} = 0.3–0.5$ (see above). To determine the half-lives of the *HTB1, HTB2,* and *ACT1* mRNAs, cells were treated with the RNA polymerase inhibitor thiolutin (Biomol) at a final concentration of 8 µg/mL (Pelechano and Pérez-Ortín, 2008). After the addition of thiolutin, 4 mL samples were taken at given time points from 0 to 60 min of incubation. Cells were harvested by centrifugation (2500 × *g*, 3.5 min) and washed with 1 mL of RNAse-free water (Qiagen) before being spun down (10k rpm, 2 min) and resuspended in 80 µL of digestion buffer (included in the YeaStar RNA Kit, Zymo Research). Cells were then stored on ice until ready for RNA extraction, which was performed using the

YeaStar RNA Kit (Zymo Research). To remove DNA contaminations, the RNA samples were treated with DNAse I (Life Technologies). Relative changes in mRNA concentrations were measured by RT-qPCR as described above. Target-specific primers were designed to bind to the *HTB1, HTB2*, and *ACT1* coding sequences, respectively (Appendix Table S4) (Bhagwat et al, 2021). The half-lives of histone and *ACT1* mRNAs were calculated by fitting a single exponential function to the obtained mRNA decay curves.

## Flow cytometry

Cells (2–5 mL) were cultured at 30 °C in different growth media for 36 h. During growth, cell cultures were kept at $OD_{600}$ <1. For each sample, cell volume distributions were measured using a Coulter Counter, and bud counts were performed to estimate the fraction of budded and unbudded cells in the populations. To analyse the fluorescence intensity of mCitrine expressed in wild-type cells, flow cytometry measurements were carried out on a 577 CytoFlex S Flow Cytometer (Beckman Coulter). mCitrine was excited with a 488-nm laser and detected using a 525/40-nm bandpass filter. 50,000 events were analysed in each experiment at a flow rate of 10 µL/min, which corresponds to roughly 1000 events/sec. Manual gating based on the side scatter (SSC) and forward scatter (FSC) parameters was performed using the FlowJo 10.8.1 software (Becton Dickinson, San Josè, CA) to eliminate doublets and cell debris. Wild-type cells not expressing mCitrine were analysed in all conditions to correct for autofluorescence.

Flow cytometry was also applied to determine the cell cycle distribution of wild-type cells in different nutrients by quantification of the DNA content. For this purpose, cells were fixed and stained with the fluorescent DNA-binding dye SYBR Green I, according to a previously published protocol (Örd et al, 2019). Briefly, 1 mL of cell culture grown for 36 h (final $OD_{600} = 0.5$) was slowly added to 9 mL of 80% ethanol and incubated overnight at 4 °C. Next, cells were pelleted (2.500 × *g*, 2 min, 4 °C) and washed twice with 50 mM Tris-HCl (pH = 8.0) before being treated with 300 µL of 1 mg/mL RNase A at 37 °C for 40 min. After washing with 50 mM Tris-HCl (pH = 8.0), cells were incubated in 50 µL 20 mg/mL Proteinase K at 37 °C for 60 min. In the last step, cells were washed again with 50 mM Tris-HCl (pH = 8.0) and treated with 200 µL of 10x SYBR Green I (Sigma-Aldrich) DNA stain at 22 °C for 1 h. For quantification of the cellular DNA content, SYBR Green I was excited with a 488-nm laser and detected using a 525/40-nm bandpass filter. The obtained DNA frequency histograms showed defined G1 and G2 peaks and were analysed with the FlowJo 10.8.1 software (Becton Dickinson, San Josè, CA). The Watson pragmatic algorithm was used to model the cell cycle and estimate the percentages of cells in the different cell cycle phases.

SYBR Green I is not suitable for staining the DNA of cells expressing mCitrine due to the overlapping emission spectra of the two fluorophores. However, cells with *mCitrine*-tagged H2B also show fluorescence histograms with distinct G1 and G2 peaks, since core histone synthesis is tightly coupled to the DNA replication during S-phase. Thus, live cells were measured as described above, and the obtained fluorescence histograms of mCitrine were analyzed in order to determine the cell cycle distributions in different nutrient conditions.

## Single-molecule fluorescence in situ hybridization (smFISH)

For smFISH analysis, commercially available Stellaris® FISH probes were designed using the Stellaris® FISH Probe Designer (Biosearch Technologies). More precisely, the *MDN1* and *mCitrine* transcripts were targeted with probe sets consisting of 27 to 48 20-mer oligonucleotides labeled with the dye Quasar-670®, while *ACT1* transcripts were bound by 41 individual Quasar-570®-labeled probes.

smFISH samples were prepared following the Stellaris® RNA FISH protocol for *S. cerevisiae*, available online at www.biosearchtech.com/stellarisprotocols. About 45 ml cultures were grown in different growth media for at least 18 h to an $OD_{600}$ = 0.3–0.5 before being fixed for 45 min at room temperature by adding formaldehyde to a final concentration of 4%. After centrifugation ($1600 \times g$, 4 min), cells were washed twice with 1 mL of ice-cold fixation buffer (1.2 M sorbitol (Sigma-Aldrich), 0.1 M $K_2HPO_4$ (Sigma-Aldrich), pH 7.5) and incubated at 30 °C for 55 min in 1 mL of fixation buffer containing 6.25 µg zymolyase (Biomol). Digested cells were then washed twice with 1 mL of ice-cold fixation buffer and stored overnight at 4 °C in 1 mL of 70% ethanol. Next, 300 uL of cells were spun down and hybridized overnight at 30 °C with 100 uL of Stellaris® RNA FISH hybridization buffer (Biosearch Technologies) containing 10% v/v formamide and 125 mM smFISH probes. The following morning, cells were washed with 10% v/v formamide in Stellaris® RNA FISH wash buffer A (Biosearch Technologies) and incubated in 1 mL of DAPI staining solution (5 ng/mL DAPI in wash buffer A with 10% v/v formamide) for 30 min at 30 °C. After washing with 1 mL of Stellaris® RNA FISH wash buffer B (Biosearch Technologies), cells were mounted in Vectashield® mounting medium (Vector Laboratories). Wide-field fluorescence imaging was performed using a Zeiss LSM 800 microscope equipped with a 63×/1.4 NA oil immersion objective and an Axiocam 506 camera. Multicolor z-stacks composed of 20 images were recorded at 240 nm intervals with the Zen 2.3 software. Quasar-570® and Quasar-670® were illuminated with a 530 nm LED and a 630 nm LED, respectively, while DAPI images were taken using a 385 nm LED.

### smFISH image analysis

Cell segmentation was performed using Cell-ACDC (Padovani et al, 2022). Briefly, cells were segmented based on a bright-field signal using YeaZ (Dietler et al, 2020) and buds were manually assigned to the correct mother cells. Cell volumes were automatically calculated by Cell-ACDC starting from the generated 2D segmentation masks. To detect and count the number of fluorescence spots in 3D, we developed a custom routine written in Python. The analysis steps are the following: (1) Application of a 3D Gaussian filter with a small sigma (0.75 voxel) to the mRNA signal. (2) Instance segmentation of the spots' signal using the best-suited automatic thresholding algorithm (either the threshold triangle, Li, or Otsu algorithms from the Python library scikit-image (van der Walt et al, 2014). (3) 3D local maxima detection (peaks) in the spots signal using the *peak_local_max* function from the Python library scikit-image. Note that peaks are searched only on the segmentation masks determined in the previous step. (4) Discarding of overlapping peaks: if two or more peaks are within a resolution-limited volume, only the peak with the highest intensity

is retained. The resolution-limited volume is determined as a spheroid with x and y radii equal to the Abbe diffraction limit and z radius equal to 1 µm. For example, with a numerical aperture of 1.4 and Quasar-670® emission wavelength of about 668 nm, the resolution-limited volume has x = y = 0.291 µm radius. (5) The remaining peaks undergo a subsequent iterative filtering routine: for each peak, we computed the Glass' delta (a measure of the effect size) of the peak's pixels compared to the background's pixels. The pixels belonging to the peaks are defined as the pixels inside the resolution-limited volume explained in step 4. The pixels belonging to the background are those pixels outside of all the detected peaks but inside the segmentation mask of the cell. The Glass's delta is computed as the mean of the peak's signal (positive signal) subtracted by the mean of the background's signal (negative or control signal) all divided by the standard deviation of the background's signal. Peaks that have an effect size lower than a threshold are discarded. The threshold value was manually determined for each experiment after careful inspection of the images. This value ranged from 0.2 to 1.0. Finally, step 5 was repeated until the number of peaks stopped changing. The final number of peaks corresponds to the detected mRNA spots.

Negative controls were performed to distinguish the mRNA signal from background noise or nonspecific probe binding. Cells carrying an extra copy of the *HTB1*, *HTB2*, or *ACT1* promoter driving mCitrine, as well as wild-type cells not expressing mCitrine, were treated with Quasar-670®-labeled probes against *mCitrine* (Fig. EV4H–J). For the analysis of *ACT1* and *MDN1* mRNA expression, wild-type cells were incubated with or without probes against *ACT1* and *MDN1* (Fig. EV4A–F). As an additional negative control, global transcription in wild-type cells was blocked by an 80-min treatment with 8 µg/mL thiolutin (Biomol) (see above) prior to incubation with *MDN1* probes (Fig. EV4G).

To determine the cell cycle stage, the cells were grouped into either G1-, S-, or G2/M-phase based on the bud-to-mother cell volume ratio. Unbudded cells containing one nucleus were classified as G1 cells. If a bud was present, the ratio of bud volume divided by mother volume was used as a classification criterion. Cells with a ratio <0.3 were considered to be in S-phase. For ratios >0.3, cells were classified as G2/M cells (Claude et al, 2021). Lastly, if cells were budded and contained two nuclei, cell outlines were carefully inspected in the bright-field images to distinguish G2/M cells from two neighboring G1 cells. mRNA concentrations were estimated as the number of mRNA spots per cell divided by the cell volume

### Live-cell fluorescence microscopy

Cells, pre-cultured in YPD, were transferred to a selected growth medium and were grown for at least 18 h to $OD_{600}$ = 0.3–0.9. For live-cell imaging, 200 µL of cells were sonicated for 5 s before being loaded into a CellASIC® ONIX2 Y04C microfluidic plate (Y04C, Millipore) connected to the ONIX2 (Millipore) microfluidic pump system. Cells were trapped inside the microfluidic chamber, and fresh medium was supplied with a constant pressure of 13.8 kPa. Live-cell fluorescence microscopy was performed on a Zeiss LSM 800 microscope equipped with an epifluorescence setup and coupled to an Axiocam 506 camera. Cells were imaged in a 30 °C incubation chamber (Incubator XLmulti S1, Pecon) at a 3 min interval over the course of 7 to 12 h using an automated stage (WSB Piezo Drive Can) and a 40×/1.3 NA oil immersion objective.

Cells with *mCitrine*-tagged *HTB1* and/or *HTB2* were illuminated for 10 ms with a 511 nm LED set at 5% power. To measure mCitrine expression from the *HTB1*, *HTB2*, or *ACT1* promoter, the LED power was increased to 12% due to the lower emission intensity of mCitrine in those cells.

For analysis, the collected microscopy images were aligned and cropped to the region of interest using a custom Fiji script (Schindelin et al, 2012). Cell segmentation and tracking were performed as previously described by Doncic et al (Doncic et al, 2013). Briefly, cells were segmented based on the phase-contrast images and tracked backward in time. Pedigrees were then manually annotated, and the time points of cell birth, bud emergence, and division were identified through visual inspection for all daughter cells born during the time-lapse experiment. The total fluorescence measured per cell was background-corrected as described by Chandler-Brown et al (Chandler-Brown et al, 2017). Moreover, for quantification of the fluorescence intensity of mCitrine expressed from the *HTB1*, *HTB2*, or *ACT1* promoter, nutrient-and cell volume-dependent autofluorescence was subtracted (Chandler-Brown et al, 2017). Autofluorescence was determined by analyzing wild-type cells not expressing a fluorescent protein.

In the case of *mCitrine*-labeled histones, autofluorescence is much weaker than the mCitrine fluorescence signal in all nutrient conditions (Appendix Fig. S1), and was therefore neglected. For the control experiments shown in Appendix Fig. S1, cell segmentation and quantification of the total fluorescence signal per cell were performed using Cell-ACDC (Padovani et al, 2022).

For all cells born during the experiment, fluorescence intensity traces were analyzed to quantify mCitrine dynamics during the cell cycle. The single-cell expression profile of the endogenously tagged histones shows a plateau during G1, followed by a twofold increase in fluorescence starting around bud emergence and a second plateau during G2/M-phase. The total amount of mCitrine produced during the cell cycle was calculated as the difference between the median of the first four time points and the median of the last four time points. To determine the duration of histone production from the single-cell traces, mean values of the G1 and G2 plateaus ($P_{G1}$, $P_{G2}$) were determined. Using a threshold of 10%, the histone production phase was then defined as the duration between the last and first time point for which $I_{mCitrine} < 1.1*P_{G1}$ and $I_{mCitrine} > 0.9*P_{G2}$, respectively.

For the nutrient upshift experiment, cells with mCitrine-tagged H2B were grown in a microfluidic plate on SCGE for 5 h before being shifted to the YPD medium. Total mCitrine intensity in newborn daughter cells was measured over the course of the experiment by time-lapse microscopy as described above. Cell segmentation and quantification of the total fluorescence signal per cell were performed using Cell-ACDC (Padovani et al, 2022), with YeaZ used for segmentation (Dietler et al, 2020).

### Protein stability measurements

To determine the stability of *HTB1-mCitrine* expressed from a β-estradiol-inducible promoter, asynchronous cell populations were cultured for at least 18 h in YPD or SCGE supplemented with 10 nM or 7 nM β-estradiol, respectively. After washing twice to remove β-estradiol from the growth medium, cells were transferred to hormone-free YPD or SCGE, and the mean *HTB1-mCitrine* fluorescence intensity was measured at defined time points by flow cytometry (see above).

### Statistical analysis

Statistical analysis was performed using GraphPad Prism 9.4.1 All datasets were tested for normality using the Shapiro–Wilk test at a confidence level of α = 0.05. Significances were calculated using an unpaired, two-tailed *t*-test for datasets that follow a Gaussian distribution or a Mann–Whitney test for datasets that are not normally distributed. To compare the slopes of two linear regression lines, *p* values were computed using GraphPad Prism 9.4.1.

## Data availability

Source data for Figs. 1–6 are provided with this paper and raw microscopy files are available at: https://www.ebi.ac.uk/biostudies/bioimages/studies/S-BIAD1308. Code for cell-ACDC is available at: https://github.com/SchmollerLab/Cell_ACDC. Custom code used to detect and count the number of fluorescence spots is available at: https://github.com/SchmollerLab/ChatziFISH

The source data of this paper are collected in the following database record: biostudies:S-SCDT-10_1038-S44318-024-00227-w.

## Peer review information

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

## Acknowledgements

We thank Christopher Bruhn, Marco Foiani, and Jennifer Ewald for sharing strains, and Pascal Falter-Braun, Christof Osman and members of the Institute of Functional Epigenetics for insightful scientific discussions. We thank Matthew Swaffer for helpful comments on the manuscript. This work was funded by the Deutsche Forschungsgemeinschaft (DFG, German Research Foundation) through projects 416098229 and 431480687, by the Human Frontier Science Program (career development award to KMS), and the Helmholtz Association.

## Author contributions

**Dimitra Chatzitheodoridou:** Conceptualization; Software; Formal analysis; Investigation; Visualization; Methodology; Writing—original draft; Writing—review and editing. **Daniela Bureik:** Conceptualization; Formal analysis; Investigation; Methodology; Writing—review and editing. **Francesco Padovani:** Software; Formal analysis; Supervision; Investigation; Visualization; Methodology; Writing—review and editing. **Kalyan V Nadimpalli:** Formal analysis; Investigation; Writing—review and editing. **Kurt M Schmoller:** Conceptualization; Resources; Software; Formal analysis; Supervision; Funding acquisition; Investigation; Methodology; Project administration; Writing—review and editing.

Source data underlying figure panels in this paper may have individual authorship assigned. Where available, figure panel/source data authorship is listed in the following database record: biostudies:S-SCDT-10_1038-S44318-024-00227-w.

## Funding

## Disclosure and competing interests statement

The authors declare no competing interests.

# Expanded View Figures

**Figure EV1. Analysis of nutrient-dependent histone H2B amounts at steady state and during nutrient upshift.**  ▶

(A) Doubling times were calculated from growth curves of wild-type cells growing in different nutrient conditions as a function of the nutrient-dependent cell volumes measured with a Coulter counter. Mean and standard deviation of $n = 4$–6 replicates are shown; line shows linear fit. (B) Representative western blot membrane stained with Ponceau S for quantification of total proteins. (C) Total protein content, extracted from equal numbers of cells in different growth media, is normalized on YPD and plotted against the nutrient-dependent cell volume measured with a Coulter counter. Mean and standard deviation of $n = 4$–7 replicates are shown; line shows linear fit. (D) Single-cell expression profiles of H2B-mCitrine (Htb1 and Htb2 tagged) corresponding to Fig. 1I ($n_{YPD} = 87$, $n_{SCD} = 83$, $n_{SCGE} = 55$). (E) Quantification of the duration of the H2B-mCitrine production phase in the different nutrients. Box plots represent the median and 25th and 75th percentiles, whiskers indicate the 5th and 95th percentiles, and symbols show outliers. The H2B-mCitrine production phase was determined as described in Methods and Materials; ***$P_{YPD-SCGE} = 0.0003$; **$p_{SCD-SCGE} = 0.0023$. (F) Total H2B-mCitrine amounts in G1 were quantified using flow cytometry in different growth media. Lines represent the mean of $n = 2$ independent replicates, each shown as an individual dot. (G) Nutrient-dependent cell cycle distributions (percentage of cells in G1-, S-, and G2/M-phase) of cells with *mCitrine*-tagged H2B, determined with flow cytometry using the H2B fluorescence intensity. The bar graphs represent the mean of $n = 2$ independent replicates, each shown as an individual dot. (H) Characterization of H2B-mCitrine strain in different growth media. Doubling times were calculated from growth curves of cells with *mCitrine*-tagged *HTB1* and *HTB2*, growing in different nutrient conditions. Lines and error bars represent the means and standard deviations of $n = 3$ independent measurements, each shown as an individual dot. (I) Nutrient-specific mean cell volumes of cells with *mCitrine*-tagged *HTB1* and *HTB2* were measured with a Coulter counter. Lines represent the mean of $n = 2$ independent measurements shown as individual dots. (J) RT-qPCR was used to measure the mRNA concentrations of *HTB1* and *HTB2*, as well as the control gene *ACT1* in wild-type cells and cells with *mCitrine*-tagged *HTB1* and *HTB2*. mRNA concentrations were normalized on *RDN18* and are shown as mean fold changes compared to the wild-type strain. Error bars indicate standard errors of at least three independent biological replicates. (K) Cells maintain constant histone protein amounts during nutrient upshift. To study how cells adjust histone production in response to a dynamic nutrient switch, cells with mCitrine-tagged H2B were grown in a microfluidic plate on SCGE for 5 h before being shifted to YPD medium (time = 0). Total mCitrine intensity in newborn daughter cells was measured over the course of the experiment using time-lapse microscopy ($n_{SCGE} = 73$, $n_{YPD} = 519$, $n_{background} = 565$). Lines connect binned means with error bars representing standard errors.

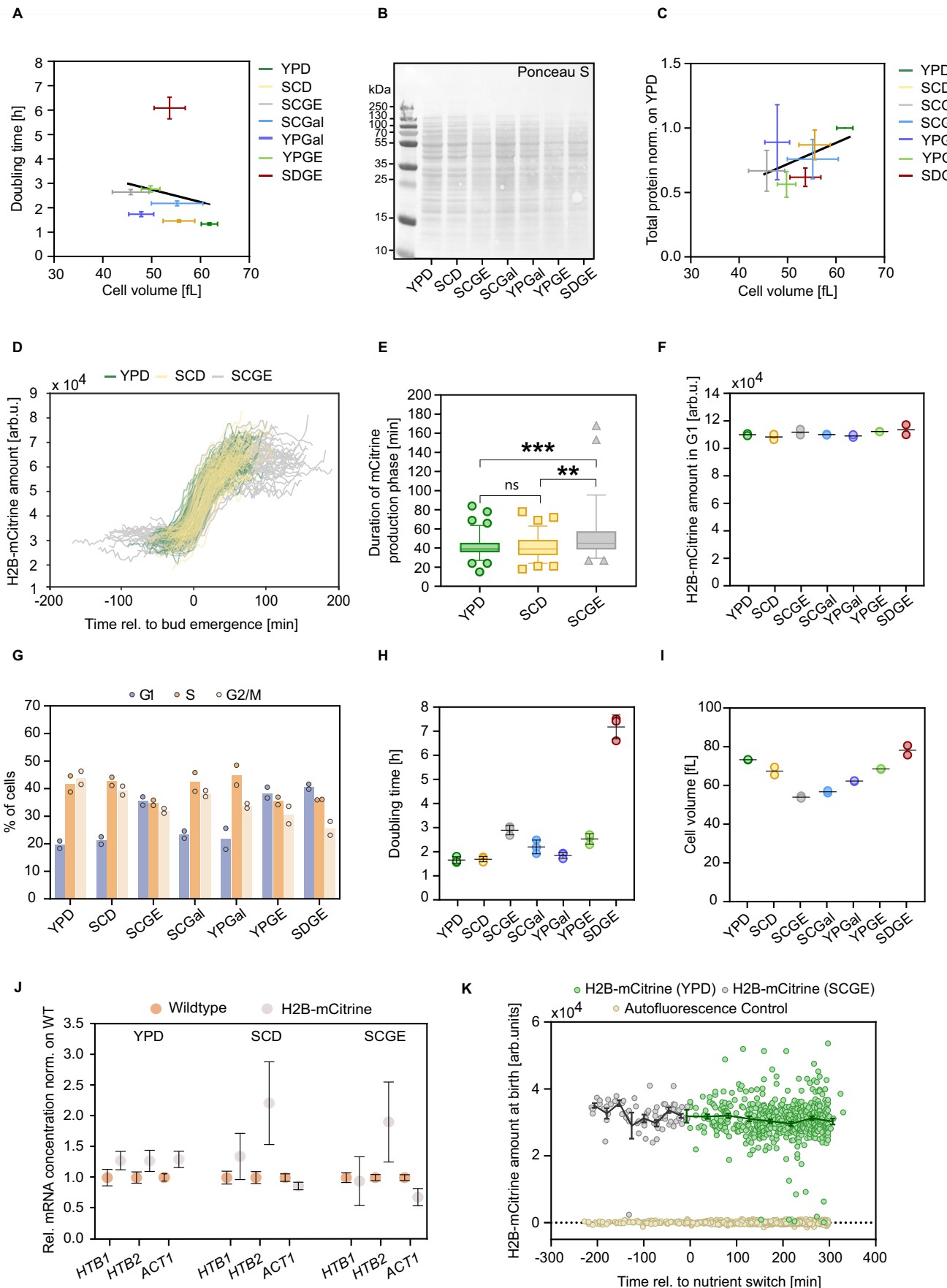

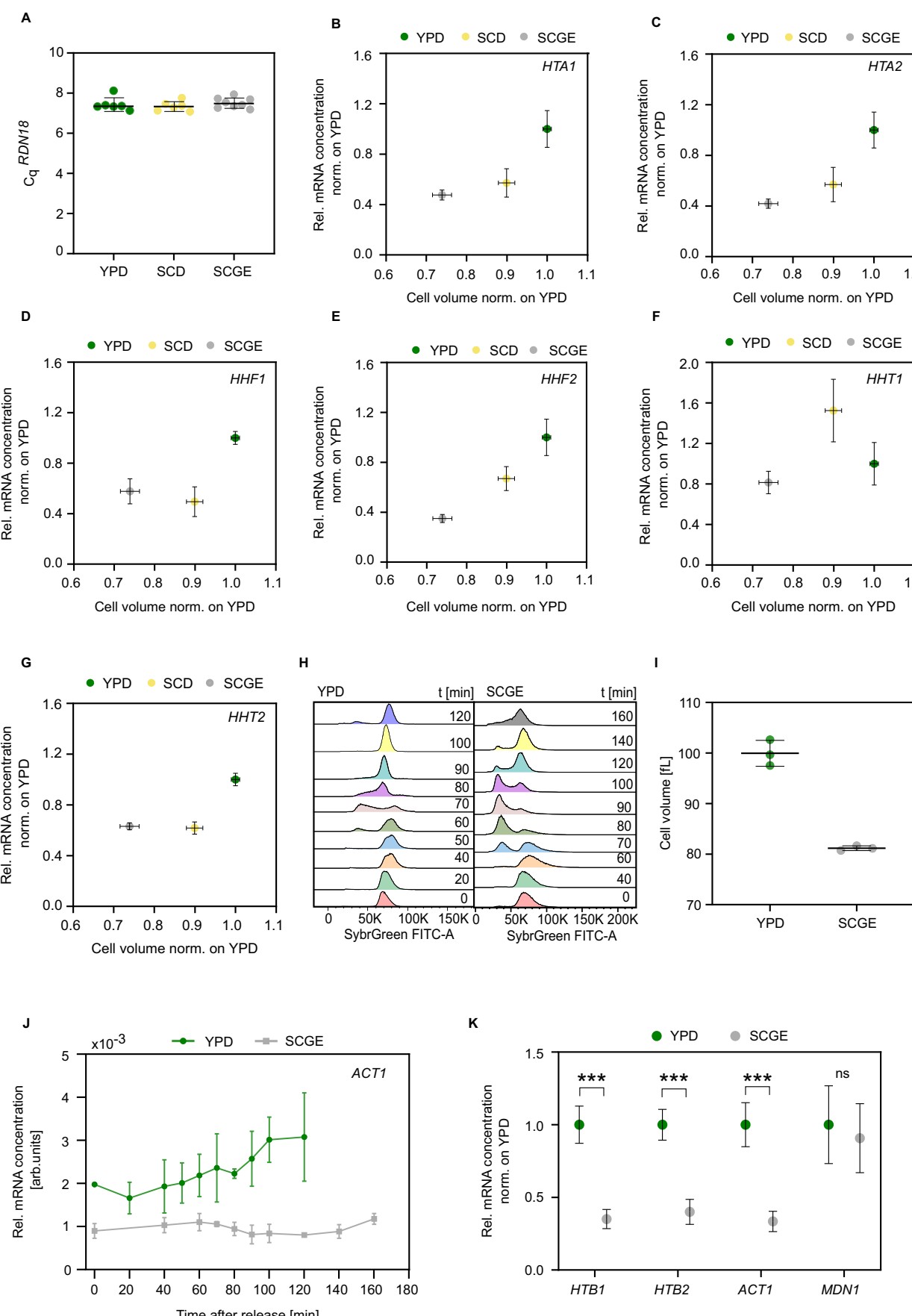

◄

**Figure EV2.  Histone mRNA concentrations decrease with decreasing nutrient-specific cell volume.**

(**A**) $C_q$ values of the reference ribosomal RNA *RDN18* were obtained by RT-qPCR analysis across different nutrient conditions. Lines and error bars represent the means and standard deviations of $n = 6–8$ independent measurements shown as individual dots. (**B–G**) Relative mRNA concentrations of *HTA1* (**B**), *HTA2* (**C**), *HHF1* (**D**), *HHF2* (**E**), *HHT1* (**F**), and *HHT2* (**G**) as a function of the relative nutrient-specific cell volume. Mean and standard deviation of at least four biological replicates are shown. (**H**) Exemplary nutrient-dependent cell cycle distributions were measured by flow cytometry at defined time points throughout the cell cycle of synchronized cells with β-estradiol-inducible *CDC20*. (**I**) Mean cell volumes of cells with β-estradiol-inducible *CDC20*, measured in YPD and SCGE at $t = 70$ min and $t = 100$ min, respectively. Lines and error bars represent the means and standard deviations of $n = 3$ independent measurements shown as individual dots. (**J**) mRNA concentrations of *ACT1* (normalized to *RDN18*) were determined using RT-qPCR after synchronous release into the cell cycle ($t = 0$) triggered by the addition of 200 nM β-estradiol. Mean and standard deviation of four biological replicates are shown. (**K**) RT-qPCR analysis of the relative mRNA concentrations of *HTB1*, *HTB2*, *ACT1*, and *MDN1* in asynchronous cells with β-estradiol-inducible *CDC20*. Mean fold changes with respect to YPD and standard deviations of $n = 3–6$ replicate measurements are shown; ***$P_{HTB1} = 1.35 \times 10^{-5}$; ***$P_{HTB2} = 2.18 \times 10^{-5}$; ***$P_{ACT1} = 3.08 \times 10^{-5}$.

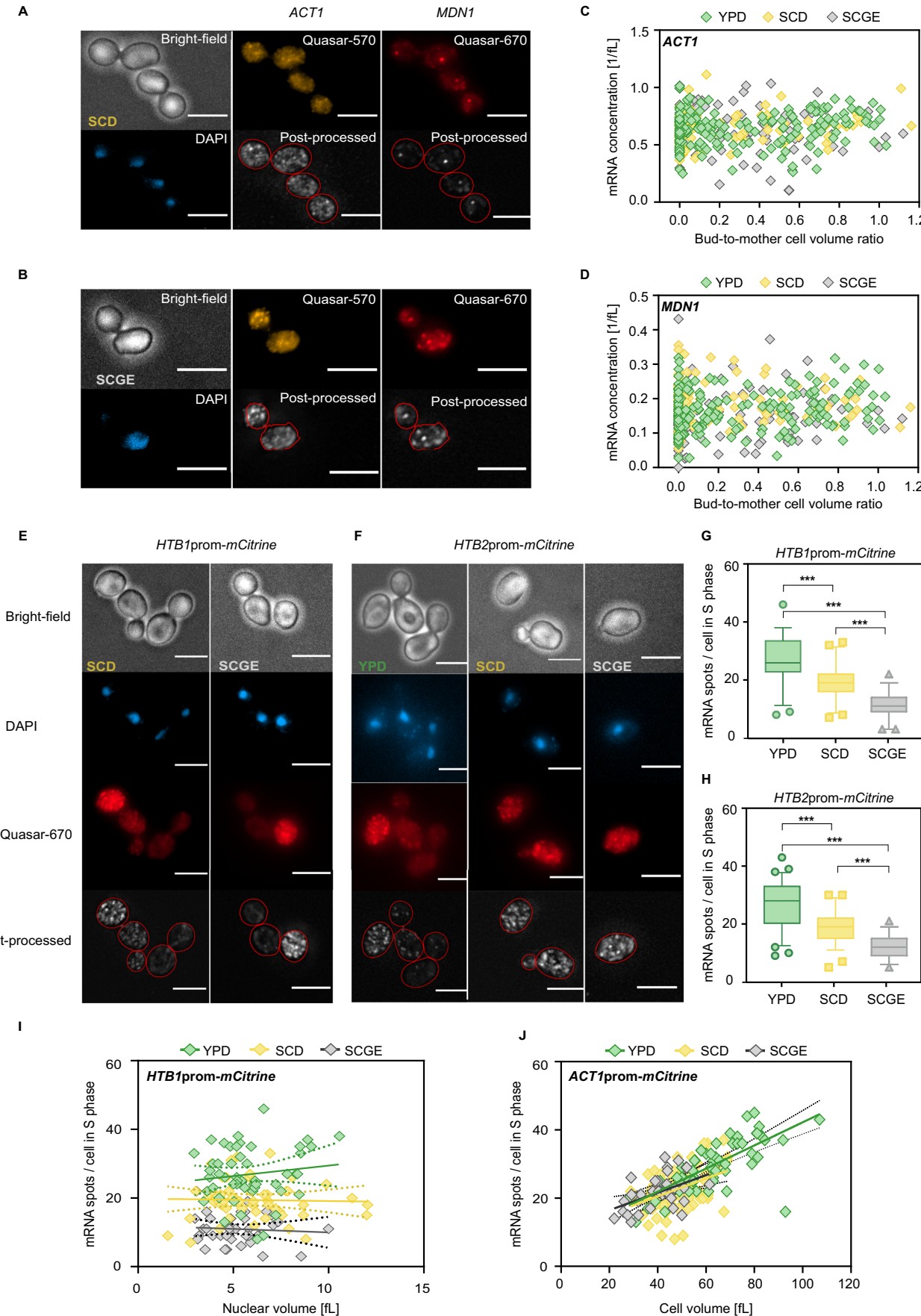

**Figure EV3.  smFISH analysis of cell cycle-dependent gene expression in different nutrient conditions.**

*ACT1* and *MDN1* transcripts were detected in single cells using smFISH probes labeled with Quasar-570 (yellow) and Quasar-670 (red), respectively. Nuclear DNA was stained with DAPI (blue). For mRNA quantification, images were post-processed as described in Methods and Materials. Representative images of cells grown in SCD (**A**) and SCGE (**B**) are shown. The scale bars represent 5 μm. (**C, D**) mRNA concentrations (mRNA spots per cell volume) of (**C**) *ACT1* ($n_{YPD} = 176$, $n_{SCD} = 87$, $n_{SCGE} = 98$) and (**D**) *MDN1* ($n_{YPD} = 176$, $n_{SCD} = 87$, $n_{SCGE} = 98$) were plotted against the corresponding bud-to-mother cell volume ratio in different nutrients. (**E, F**) Representative images of cells expressing (**E**) *HTB1prom-mCitrine* and (**F**) *HTB2prom-mCitrine* in different growth media. *mCitrine* transcripts were detected using probes labeled with Quasar-670 (red). Cell nuclei were stained with DAPI (blue). For mRNA quantification, images were post-processed as described in *Methods and Materials*. Scale bars represent 5 μm. (**G, H**) *mCitrine* mRNA amounts expressed from the (**G**) *HTB1* (*HTB1prom-mCitrine*; $n_{YPD} = 50$, $n_{SCD} = 51$, $n_{SCGE} = 39$) and (**H**) *HTB2* promoter (*HTB2prom-mCitrine*; $n_{YPD} = 64$, $n_{SCD} = 59$, $n_{SCGE} = 50$) in S-phase, as quantified by smFISH. Box plots represent median and 25th and 75th percentiles, whiskers indicate the 5th and 95th percentiles and symbols show outliers; *HTB1prom-mCitrine* (***$P_{YPD-SCGE} = 5.29 \times 10^{-20}$; ***$P_{YPD-SCD} = 4.07 \times 10^{-7}$; ***$P_{SCD-SCGE} = 2.31 \times 10^{-11}$); *HTB2prom-mCitrine* (***$P_{YPD-SCGE} = 5.42 \times 10^{-23}$; ***$P_{YPD-SCD} = 1.19 \times 10^{-8}$; ***$P_{SCD-SCGE} = 4.63 \times 10^{-11}$). (**I**) Number of *HTB1prom-mCitrine* mRNA spots per cell ($n_{YPD} = 50$, $n_{SCD} = 51$, $n_{SCGE} = 39$) as a function of nuclear volume during S-phase. Here, cells with a bud-to-mother volume ratio <0.3 were considered to be in S-phase. Lines show linear fits; dashed lines indicate the 95% confidence intervals. (**J**) Number of *ACT1* promoter-*mCitrine* mRNA spots per cell ($n_{YPD} = 59$, $n_{SCD} = 75$, $n_{SCGE} = 33$) as a function of cell volume during S-phase. Here, cells with a bud-to-mother volume ratio <0.3 were considered to be in S-phase. Lines show linear fits; dashed lines indicate the 95% confidence intervals.

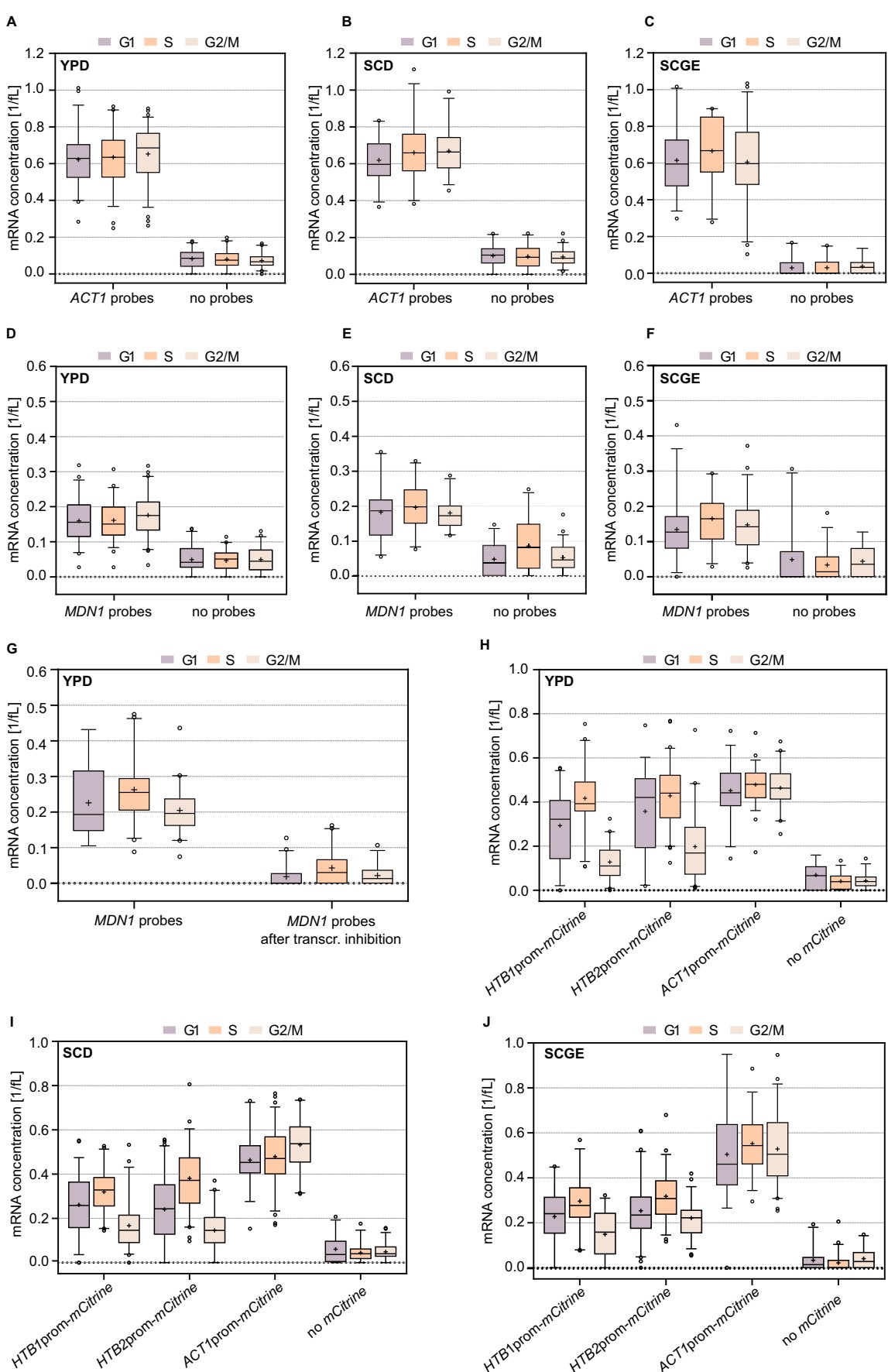

◀ **Figure EV4.  mRNA concentration of transcripts of interest in G1-, S- and G2/M-phase measured with smFISH are shown in comparison to negative controls.**

(A–F) Wild-type cells were incubated with or without smFISH probes against *ACT1* (A–C) or *MDN1* (D–F), *ACT1, MDN1* ($n_{G1}^{YPD} = 48$, $n_{S}^{YPD} = 54$, $n_{G2M}^{YPD} = 74$, $n_{G1}^{SCD} = 26$, $n_{S}^{SCD} = 31$, $n_{G2M}^{SCD} = 30$, $n_{G1}^{SCGE} = 26$, $n_{S}^{SCGE} = 23$, and $n_{G2M}^{SCGE} = 49$); no probes ($n_{G1}^{YPD} = 49$, $n_{S}^{YPD} = 40$, $n_{G2M}^{YPD} = 43$, $n_{G1}^{SCD} = 28$, $n_{S}^{SCD} = 27$, $n_{G2M}^{SCD} = 52$, $n_{G1}^{SCGE} = 24$, $n_{S}^{SCGE} = 30$, and $n_{G2M}^{SCGE} = 16$). (G) Wild-type cells were treated with thiolutin for 80 min to inhibit global transcription prior to incubation with *MDN1* probes. mRNA concentrations in G1, S, and G2/M were estimated by dividing the number of detected spots by the cell volume ($n_{G1}^{MDN1} = 14$, $n_{S}^{MDN1} = 42$, $n_{G2M}^{MDN1} = 40$, $n_{G1}^{MDN1,aftertranscr.inhibition} = 57$, $n_{S}^{MDN1,aftertranscr.inhibition} = 40$, and $n_{G2M}^{MDN1,aftertranscr.inhibition} = 34$). (H–J) Wild-type cells expressing no mCitrine, as well as cells carrying an additional copy of the *HTB1, HTB2*, or *ACT1* promoter driving mCitrine were incubated with smFISH probes against *mCitrine*. mRNA concentrations in G1, S, and G2/M were estimated by dividing the number of detected spots by the cell volume. Box plots represent median and 25th and 75th percentiles; whiskers indicate the 5th and 95th percentiles and symbols show outliers; *HTB1prom-mCitrine* ($n_{G1}^{YPD} = 58$, $n_{S}^{YPD} = 49$, $n_{G2M}^{YPD} = 41$, $n_{G1}^{SCD} = 54$, $n_{S}^{SCD} = 51$, $n_{G2M}^{SCD} = 53$, $n_{G1}^{SCGE} = 28$, $n_{S}^{SCGE} = 39$, and $n_{G2M}^{SCGE} = 28$); *HTB2prom-mCitrine* ($n_{G1}^{YPD} = 37$, $n_{S}^{YPD} = 64$, $n_{G2M}^{YPD} = 59$, $n_{G1}^{SCD} = 85$, $n_{S}^{SCD} = 59$, $n_{G2M}^{SCD} = 50$, $n_{G1}^{SCGE} = 65$, $n_{S}^{SCGE} = 50$, and $n_{G2M}^{SCGE} = 55$); *ACT1prom-mCitrine* ($n_{G1}^{YPD} = 35$, $n_{S}^{YPD} = 59$, $n_{G2M}^{YPD} = 41$, $n_{G1}^{SCD} = 38$, $n_{S}^{SCD} = 75$, $n_{G2M}^{SCD} = 42$, $n_{G1}^{SCGE} = 37$, $n_{S}^{SCGE} = 33$, and $n_{G2M}^{SCGE} = 54$); *no mCitrine* ($n_{G1}^{YPD} = 15$, $n_{S}^{YPD} = 28$, $n_{G2M}^{YPD} = 37$, $n_{G1}^{SCD} = 32$, $n_{S}^{SCD} = 30$, $n_{G2M}^{SCD} = 48$, $n_{G1}^{SCGE} = 26$, $n_{S}^{SCGE} = 49$, and $n_{G2M}^{SCGE} = 31$).

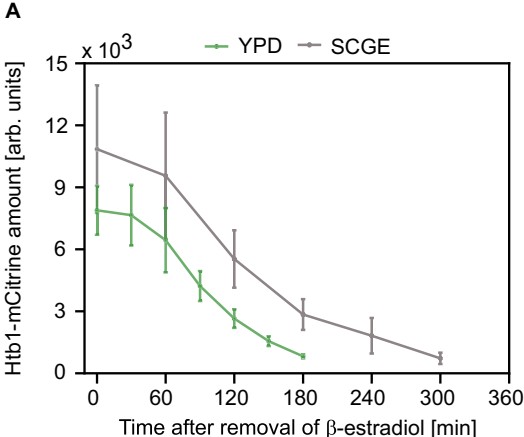
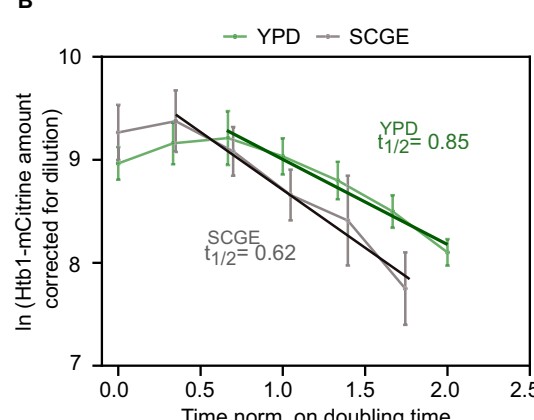

**Figure EV5.  Analysis of nutrient-dependent histone protein degradation suggests that histone stability does not compensate for the transcriptional downregulation in poor growth media.**

To study the degradation of histone proteins in different nutrients, we constructed a strain carrying an extra copy of *HTB1-mCitrine* expressed from a β-estradiol-inducible promoter. First, Htb1-mCitrine was expressed by the addition of 10 nM or 7 nM β-estradiol to YPD or SCGE, respectively. Upon removal of the hormone from the growth media, protein synthesis was inhibited, and the degradation of Htb1-mCitrine in asynchronous cell cultures was monitored over time using flow cytometry. (**A**) Htb1-mCitrine protein degradation curves in YPD and SCGE measured after β-estradiol removal. For each time point, the mean and standard deviation of three biological replicates is plotted. (**B**) Nutrient-specific protein half-lives were obtained from linear regression on the natural logarithm of the mCitrine amounts as a function of time normalized to the respective doubling time. The mCitrine amounts were corrected to account for the dilution through cell division. For each time point, the mean and standard deviation of three biological replicates is shown.

