## [Peer Review File · The EMBO Journal]

Decoupled transcript and protein concentrations ensure histone homeostasis in different nutrients

Dimitra Chatzitheodoridou, Daniela Bureik, Francesco Padovani, Kalyan Nadimpalli, and Kurt Schmoller

Corresponding author(s): Kurt Schmoller (kurt.schmoller@helmholtz-munich.de)

Review Timeline:

Submission Date:	7th Feb 24
Editorial Decision:	9th Apr 24
Revision Received:	3rd Jun 24
Editorial Decision:	10th Jul 24
Revision Received:	5th Aug 24
Accepted:	27th Aug 24

Editor: Hartmut Vodermaier

Transaction Report:

Dear Dr. Schmoller,

Thank you for submitting your manuscript on histone protein homeostasis across various growth rates for our consideration. We have now received reports from three expert referees, which I am passing on below for your information. As you will see, the referees find the results of your investigation interesting in principle, but also point out that it remains unclear which factors/mRNA features would govern nutrient-specific translation, and how this would ensure constant histone levels. Since it is not clear if this key issue (and various other specific concerns) would be readily addressable during a regular, single-round revision, I would be interested in hearing how you would tackle the referees' comments, should you be given the opportunity to revise this work for The EMBO Journal. Based on such a revision proposal and preliminary point-by-point response to the included reports, I could then determine whether a major revision for The EMBO Journal would seem realistic, or whether a less substantively revised version might alternatively be suitable for one of our sister journals. I'd be happy to talk through such a revision proposal with you if needed.

Looking forward to hearing from you.

Best regards,

Hartmut

Referee #1 (Report for Author)

This is an interesting paper on an intriguing topic: how histone levels are kept constant (because they have to bind the same amount of DNA) in different nutrients. Unfortunately, it falls short of offering a convincing explanation. After their initial observations, they go on to nicely summarize their results for the constancy of histone levels in different nutrients as follows: "The apparent decoupling of histone protein and mRNA abundance suggests that medium-specific regulation of protein translation or stability is required to ultimately ensure histone homeostasis in different nutritional environments." However, they devote the rest of the paper to evaluating histone transcription in various settings instead of offering evidence about how the levels of these proteins are adjusted to where they need to be. This is bizarre and totally anticlimactic, detracting from the paper's significance. Every reader would like to know how histone protein amounts are maintained across nutrient conditions. That is the critical point, in my opinion. Not how histone mRNAs are downregulated in particular nutrients. Even their model in Figure 7 highlights "nutrient-specific translation" as the driver of the constant histone levels. Yet, their experiments do not focus on translation.

ADDITIONAL POINTS

- Despite their claim that "our data suggest that to compensate for the decreased transcript concentrations in poor nutrients, the relative translation efficiency of core histones is higher than in rich nutrients" (earlier in the text and the Discussion), they do not ever measure the translational efficiency of these mRNAs in different nutrients or the stability of the proteins. They rely exclusively on the single-cell fluorescence measurements of tagged proteins as a proxy. However, this does not measure translational efficiency. For any claims about translational efficiency, they would need to measure the ribosome-bound histone mRNAs and compare it to their total mRNA abundance. These are straightforward experiments. At a minimum, they should be performed to account for their observations. Ideally, a mechanism to explain the upregulation at the protein level should also be offered.

- In contrast to their statements that cells in poor medium grow slower and are smaller, their data shows a complex relation between doubling time and cell size (compare Figure 1A with 1B). For example, YPGal cells grow faster than YPGE cells but are smaller, (with several other such comparisons). Likewise, although SDGE cells have a >2-fold longer doubling time than YPGE cells, their %G1 is the same or lower than YPGE cells. How do they explain these observations?

- In Figure 1D, they argue that cells in poor nutrients have less protein from Ponceau-stained gels. The quantitative nature of these measurements is highly questionable (also, there are no error bars for the reference YPD samples in Figure 1E), but even with that assay, is there any difference among SCD, SCGE, SCGal, and YPGal, despite their different growth rates?

- Comparing Figures 1F (relative H2B levels) to Figure 1G (relative Actin levels), they derive a central conclusion that "the histone H2B protein concentration decreases in inverse proportion with cell volume". However, much of the purported difference arises from a single outlier data point (YPGal). If one ignores that data point, the two graphs look very similar. This, along with the Ponceau approach to quantifying proteins, and immunoblots for H2B and actin, weakens these conclusions. A different, more quantitative method (histones and actin are abundant; why not use mass spectrometry-based methods?) should be added to the analysis.

- Their fluorescence experiments in Figure 1H-K are better and support the conclusion that "histone protein amounts are maintained constant across nutrient conditions." However, the way the data is presented, with emphasis on Figure 1J, showing the three nutrients they examined and histone levels in the cell cycle related to cell volume at birth, confuses the issue. Even the section title "In different growth conditions, histone protein concentrations decrease with cell volume on a single cell level" distracts and confuses unnecessarily.

- The cell cycle expression patterns in Figure 2 were obtained from an arrest (in M) and release. Then, they observe lower peak expression in the subsequent S phase in poor nutrients. The problem is that the cells continue to grow during the arrest, possibly confounding the outcome. They should have followed the second cycle to exclude arrest-related artifacts.

Referee #2 (Report for Author)

In this manuscript, the authors investigate mechanisms of histone protein homeostasis in yeast cells growing under various nutrient conditions. A key finding is that histone mRNA expression changes in these different conditions, whereas protein levels do not change. They test hypotheses for how this regulation could work and conclude that the translation efficiency of the histone mRNA is increased in nutrient-rich conditions. Overall, the experiments are carefully done and the results are interesting. Although they don't identify factors or mRNA features that influence translational efficiency, these experiments will lay a foundation for future work. I support publication once the following issues have been addressed:

1. In the second paragraph of the Results section, the authors state: "we observe a decrease in total protein abundance in nutrient-poor conditions". Many readers will require more explanation here. As written, this could be interpreted to mean that protein concentration within cells decreases. However, since the same number of cells were used the decrease in protein abundance likely reflects the fact that cells are smaller in nutrient-poor conditions, but with similar protein concentration. It is also stated in this paragraph that "the histone H2B protein concentration decreases in inverse proportion with cell volume". For many readers, it may be more simple to state that the amount of histone protein per cell stays constant across different nutrient conditions that influence cell size.
2. In Figure 2A the authors use qPCR to quantify histone mRNA levels in different media. Using qPCR in these kinds of situations can be challenging because the results can depend strongly on the reference RNA used for normalization, which could be undergoing its own changes in the different conditions that would strongly influence the results. In Figure 2A, they use total RNA as the reference. Since the vast majority of total RNA is ribosomal RNA, they are essentially normalizing to rRNA. In Figure 2C they use RDN18 to normalize. I didn't know what this is but looked it up and found that it is one of the ribosomal RNAs. The authors should edit to state that they are normalizing to rRNA in both cases. Importantly, they should include a comment or new data to describe how rRNA levels change in response to the different media, which will help readers interpret the data. This kind of information about the RNAs used for normalization should be included throughout wherever it is possible, and the authors should take extra care and explain their efforts to be sure that differences in histone mRNA levels between different media are not a qPCR artifact.
3. The authors use a *sml1Δ rad53Δ spt21Δ* strain to investigate the effects of histone overexpression and conclude that cells in nutrient-poor conditions are more vulnerable to histone overexpression, based on analysis of the growth rate of strains in rich or poor media. However, the deleted genes have many other functions in the cells, and likely have additional unknown functions that have not yet been discovered. Thus, it is difficult to conclude that the effects of the mutants are due solely to effects on histone expression. The authors should acknowledge this caveat and use appropriately cautious wording. It seems that a more rigorous approach would be to increase the copy number of histone genes. Has this been tried? Is there a mechanism that reduces histone protein levels in response to increased copy number? It would be helpful if the authors discussed these issues.

4. The mechanism by which histone protein levels are adjusted in response to reduced transcript levels in poor carbon remain unknown. The author's experiments suggest that the 5' UTR is not required. They conclude that the translation efficiency of histone mRNAs is increased in nutrient-rich conditions, and that an "imprinting" mechanism is used to add factors that control mRNA translation during the transcription process. These are interesting ideas, although not a lot of data to support them, so the authors should use cautious wording that includes a thorough discussion of where the gaps in knowledge are.

Signed review: Doug Kellogg

Referee #3 (Report for Author)

Building on previous work having shown that unlike most other proteins, histone concentration subscales with cell volume to maintain constant histone-to-DNA levels, the authors show that volume-independent de-coupled transcription and translation of histone genes contribute to homeostasis in different nutrient conditions. The authors show that, surprisingly, histone mRNA concentration is lower in poor nutrients due to regulation at the promoter level, suggesting that an increase in translation efficiency occurs in poor nutrients. The authors also offer an interesting evolutionary explanation for this phenomenon.

The paper is well written, the experiments are well thought and the data supports the conclusions well. I particularly enjoyed the clarity of the discussion. This study describes an interesting phenomenon and provides plausible hypothesis to further investigate the molecular mechanisms responsible for it.

Minor comments:

1 - Fig. 1D should be moved to the supplementary Figure 1.

2 - Writing about Fig. 1F, the authors state that "the histone H2B protein concentration decreases in inverse proportion with cell volume". The Fig. 1F data does not strongly support a proportional relationship between the variables, for that the exponent on the fitted regression should be -1 whereas it is closer to -1.5 on the graph. It is striking for instance that the YPGal population has a H2B expression more than twice the YPD population despite cells only being less than 25% less voluminous than in the YPD population and having the same cell-cycle phases distribution. Is there a technical bias in the H2B expression quantification or do the authors think that the relationship between the two variables is not strictly proportional?

3 - The y-axis label of Fig. 1G should read "Actin" instead of "H2B".

4 - On Fig. 2A, the mRNA concentrations for HTB1 and HTB2 are lower in SCD than in YPD, however on Fig. 3B the mRNA concentration under the same promoters are similar in both media (if not slightly higher in SCD). Does that imply that, at least in SCD vs YPD, the promoter regulation is not sufficient to explain the differences in mRNA concentrations of HTB1 and HTB2?

Dear Hartmut,

Thank you again for your feedback. Please find attached a point-by-point response on how we envision to address the reviewer's comments. As you will see, most of the comments can be easily addressed.

We agree that the big open question following from our work is how translation is regulated to compensate for transcriptional regulation and ensure constant histone amounts. We think the best strategy to obtain additional mechanistic insight within the time frame of a revision is to study selected deletion mutants. In particular, we would study deletions of Spt21 and Hir1, both of which are involved in the regulation of histone transcription by interacting with histone promoters. In addition, we would focus on Tor1 and Gcn2, which regulate protein synthesis according to nutrient availability. Using RT-qPCR and flow cytometry, we will examine how cells lacking one of these four factors regulate histone mRNA and protein levels in rich and poor media. In fact, our recent preliminary data indicate that deletion of Tor1 affects histone homeostasis.

We are looking forward to hearing your opinion and are happy to talk about it in a phone or video call.

Best,
Kurt

Referee #1 (Report for Author)

This is an interesting paper on an intriguing topic: how histone levels are kept constant (because they have to bind the same amount of DNA) in different nutrients. Unfortunately, it falls short of offering a convincing explanation. After their initial observations, they go on to nicely summarize their results for the constancy of histone levels in different nutrients as follows: "The apparent decoupling of histone protein and mRNA abundance suggests that medium-specific regulation of protein translation or stability is required to ultimately ensure histone homeostasis in different nutritional environments." However, they devote the rest of the paper to evaluating histone transcription in various settings instead of offering evidence about how the levels of these proteins are adjusted to where they need to be. This is bizarre and totally anticlimactic, detracting from the paper's significance. Every reader would like to know how histone protein amounts are maintained across nutrient conditions. That is the critical point, in my opinion. Not how histone mRNAs are downregulated in particular nutrients. Even their model in Figure 7 highlights "nutrient-specific translation" as the driver of the constant histone levels. Yet, their experiments do not focus on translation.

We agree with the reviewer that understanding the regulation of histone translation is now the big open question. However, we want to point out that many of the most important findings of our manuscript follow only after the quoted statement. In particular, we only afterwards showed that it is translation rather than protein degradation that is responsible for the decoupling, and that the decoupling is mediated by the untranscribed part of the promoter. These are critical insights towards understanding the decoupling, and we think that without this part of the paper, the obvious hypothesis would have been that the decoupling is simply mediated by protein degradation.

ADDITIONAL POINTS

- Despite their claim that "our data suggest that to compensate for the decreased transcript concentrations in poor nutrients, the relative translation efficiency of core histones is higher than in rich nutrients" (earlier in the text and the Discussion), they do not ever measure the translational efficiency of these mRNAs in different nutrients or the stability of the proteins. They rely exclusively on the single-cell fluorescence measurements of tagged proteins as a proxy. However, this does not measure translational efficiency. For any claims about translational efficiency, they would need to measure the ribosome-bound histone mRNAs and compare it to their total mRNA abundance. These are straightforward experiments. At a minimum, they should be performed to account for their observations. Ideally, a mechanism to explain the upregulation at the protein level should also be offered.

First, we would like to point out that in fact we measured protein stability (Supplementary Fig. 11), and we found that protein degradation does not compensate for the downregulation of histone transcripts in poor nutrients. These experiments, together with the single-cell measurements of mCitrine driven by the H2B promoters (Fig. 5A-C) leave translation as the only possible process that ensures constant histone protein amounts despite the lower transcript amounts, and we therefore think our data support our claims.

While we do not think that measurement of the ribosome-bound fraction of histone mRNA is necessary for the claims we made in our manuscript, we agree with the reviewer that such experiments might provide interesting insights, and we were planning to go in this direction in the future. At the moment, our group does not have experience with these assays, and we are therefore not sure how straightforward and quick it would be in practice to perform these experiments with the appropriate quality. However, if you think it is necessary, we would try to perform polysome profiling experiments with the help of collaborators.

- In contrast to their statements that cells in poor medium grow slower and are smaller, their data shows a complex relation between doubling time and cell size (compare Figure 1A with 1B). For example, YPGal cells grow faster than YPGE cells but are smaller, (with several other such comparisons). Likewise, although SDGE cells have a >2-fold longer doubling time than YPGE cells, their %G1 is the same or lower than YPGE cells. How do they explain these observations?

The reviewer is right that while it is a commonly stated trend that poor growth media lead to smaller cell sizes, in reality the relationship is complex. Importantly, we measure growth rate and cell size in each condition, and our conclusions do not depend on a dependence of cell size on growth rate. We will clarify this in the revised manuscript.

- In Figure 1D, they argue that cells in poor nutrients have less protein from Ponceau-stained gels. The quantitative nature of these measurements is highly questionable (also, there are no error bars for the reference YPD samples in Figure 1E), but even with that assay, is there any difference among SCD, SCGE, SCGal, and YPGal, despite their different growth rates?

The error bars in Fig. 1E show standard deviations between replicates. Because absolute quantification is not directly comparable between replicates, we first normalized on YPD for each individual replicate. By definition, YPD then shows the value 1 in Fig. 1E, without variability between replicates that could be shown as standard deviation.

While specific regulation depending on the exact media is not unlikely, we observed the expected general trend that large cells contain more protein. In the revised manuscript, we will include plots showing the relationship between i) cell size and doubling time and ii) total protein and cell size to address this point and the one raised above.

- Comparing Figures 1F (relative H2B levels) to Figure 1G (relative Actin levels), they derive a central conclusion that "the histone H2B protein concentration decreases in inverse proportion with cell volume". However, much of the purported difference arises from a single outlier data point (YPGal). If one ignores that data point, the two graphs look very similar. This, along with the Ponceau approach to quantifying proteins, and immunoblots for H2B and actin, weakens these conclusions. A different, more quantitative method (histones and actin are abundant; why not use mass spectrometry-based methods?) should be added to the analysis.

In response to the reviewer's remark, we performed a new analysis without the data obtained for YPGal to assess the impact of this apparent outlier on the observed trend. As shown in Fig. R1, even without YPGal, our results suggest an inverse relationship between H2B protein concentration and cell volume. Interestingly, without the outlier, the relationship is even closer to the inverse proportionality that is expected for constant histone amounts.

Figure R1. Western blot analysis reveals that histone protein concentrations decrease with cell volume across nutrient conditions. Mean and standard deviation of at least 3 biological replicates are shown.

Still, we agree with the reviewer that western blots are not the most quantitative method. This is why we verified our finding with flow cytometry and fluorescence microscopy. Importantly, both these single cell methods also allow us to specifically measure histone amounts of G1 cells. This is important due to the cell cycle dependence of histone expression, which implies that in bulk experiments, the measured histone amounts also depend on the cell cycle distribution. All our measurements consistently – and following the obvious expectation – show that histone amounts at a given cell cycle stage are constant. We therefore do not think that bulk mass spectrometry experiments are necessary to make this point.

- Their fluorescence experiments in Figure 1H-K are better and support the conclusion that "histone protein amounts are maintained constant across nutrient conditions." However, the way the data is presented, with emphasis on Figure 1J, showing the three nutrients they examined and histone levels in the cell cycle related to cell volume at birth, confuses the issue. Even the section title "In different growth conditions, histone protein concentrations decrease with cell volume on a single cell level" distracts and confuses unnecessarily.

We will further clarify this data presentation in the revised manuscript.

- The cell cycle expression patterns in Figure 2 were obtained from an arrest (in M) and release. Then, they observe lower peak expression in the subsequent S phase in poor nutrients. The problem is that the cells continue to grow during the arrest, possibly confounding the outcome. They should have followed the second cycle to exclude arrest-related artefacts.

To exclude a possible effect of the nutrient-dependent cell volume on histone mRNA expression after cell cycle release, we compared the corresponding cell volumes at peak expression. As shown in Figure R2, cells are larger in YPD than SCGE. Thus, the lower peak expression observed in SCGE is not a result of relatively increased cell volume.

Figure R2. Coulter counter measurements of cell volume at peak expression in YPD and SCGE. Lines indicate the mean across $n = 3$ replicate measurements, each shown as an individual dot.

Referee #2 (Report for Author)

In this manuscript, the authors investigate mechanisms of histone protein homeostasis in yeast cells growing under various nutrient conditions. A key finding is that histone mRNA expression changes in these different conditions, whereas protein levels do not change. They test hypotheses for how this regulation could work and conclude that the translation efficiency of the histone mRNA is increased in nutrient-rich conditions. Overall, the experiments are carefully done and the results are interesting. Although they don't identify factors or mRNA features that influence translational efficiency, these experiments will lay a foundation for future work. I support publication once the following issues have been addressed:

1. In the second paragraph of the Results section, the authors state: "we observe a decrease in total protein abundance in nutrient-poor conditions". Many readers will require more explanation here. As written, this could be interpreted to mean that protein concentration within cells decreases. However, since the same number of cells were used the decrease in protein abundance likely reflects the fact that cells are smaller in nutrient-poor conditions, but with similar protein concentration. It is also stated in this paragraph that "the histone H2B protein concentration decreases in inverse proportion with cell volume". For many readers, it may be more simple to state that the amount of histone protein per cell stays constant across different nutrient conditions that influence cell size.

In the revised version of the manuscript, we will further clarify these points.

2. In Figure 2A the authors use qPCR to quantify histone mRNA levels in different media. Using qPCR in these kinds of situations can be challenging because the results can depend strongly on the reference RNA used for normalization, which could be undergoing its own changes in the different conditions that would strongly influence the results. In Figure 2A, they use total RNA as the reference. Since the vast majority of total RNA is ribosomal RNA, they are essentially normalizing to rRNA. In Figure 2C they use RDN18 to normalize. I didn't know what this is but looked it up and found that it is one of the ribosomal RNAs. The authors should edit to state that they are normalizing to rRNA in both cases. Importantly, they should include a comment or new data to describe how rRNA levels change in response to the different media, which will help readers interpret the data. This kind of information about the RNAs used for normalization should be included throughout wherever it is possible, and the authors should take extra care and explain their efforts to be sure that differences in histone mRNA levels between different media are not a qPCR artifact.

The reviewer is correct. Our reasoning was that rRNA constitutes the majority of total RNA. Thus, normalization on rRNA effectively mimics normalization on total RNA. In the revised version of the manuscript, we will present evidence that the ratio of *RDN18* RNA to total RNA is largely constant between the media, and we will clarify this in the text.

3. The authors use a *sml1Δ rad53Δ spt21Δ* strain investigate the effects of histone overexpression and conclude that cells in nutrient-poor conditions are more vulnerable to histone overexpression, based on analysis of the growth rate of strains in rich or poor media. However, the deleted genes have many other functions in the cells, and likely have additional unknown functions that have not yet been discovered. Thus, it is difficult to conclude that the effects of the mutants are due solely to effects on histone expression. The authors should acknowledge this caveat and use appropriately cautious wording. It seems that a more rigorous approach would be to increase the copy number of histone genes. Has this been tried? Is there a mechanism the reduces histone protein levels in response to increased copy number? It would be helpful if the authors discussed these issues.

Histone homeostasis depends on multiple layers of regulation. In particular, a subset of genes shows transcriptional dosage compensation, and excess histone proteins are degraded in a Rad53-dependent manner (Eriksson et al., 2012). This is why instead of simply integrating additional gene copies, we chose the strategy of deleting Rad53 – which perturbs both the transcriptional and protein-degradation based control.

The reviewer is right that especially *rad53* deletion will have additional effects besides histone overexpression. In our manuscript, we followed the reasoning of Bruhn et al. that the fact that the phenotype is (partially) rescued by additional deletion of Spt21 points to a histone-specific effect. However, we agree with the reviewer, and we will discuss this caveat in the revised version of the manuscript.

4. The mechanism by which histone protein levels are adjusted in response to reduced transcript levels in poor carbon remain unknown. The author's experiments suggest that the 5' UTR is not required. They conclude that the translation efficiency of histone mRNAs is increased in nutrient-rich conditions, and that an "imprinting" mechanism is used to add factors that control mRNA translation during the transcription process. These are interesting ideas, although not a lot of data to support them, so the authors should use cautious wording that includes a thorough discussion of where the gaps in knowledge are.

We agree with the reviewer that further research is needed to unravel a potential imprinting mechanism underlying nutrient-dependent histone homeostasis. To gain new mechanistic insight, we will investigate the impact of four carefully selected factors, Spt21, Hir1, Tor1 and Gcn2, on the regulation of histones across nutrients. Spt21 and Hir1 are involved in the activation and repression of histone gene regulation, respectively, by interacting with histone promoters. Tor1 and Gcn2 coordinate protein synthesis in response to nutrient availability. Using RT-qPCR and flow cytometry, we will examine how cells lacking one of these four factors regulate histone mRNA and protein levels in rich and poor medium. In fact, our recent preliminary data indeed suggest a role for Tor1 in histone homeostasis.

According to the reviewer's request, we will also provide a more elaborate discussion and expand on our current understanding of the regulation of histone gene expression across different nutrients.

Signed review: Doug Kellogg

Referee #3 (Report for Author)

Building on previous work having shown that unlike most other proteins, histone concentration sub-scales with cell volume to maintain constant histone-to-DNA levels, the authors show that volume-independent de-coupled transcription and translation of histone genes contribute to homeostasis in different nutrient conditions. The authors

show that, surprisingly, histone mRNA concentration is lower in poor nutrients due to regulation at the promoter level, suggesting that an increase in translation efficiency occurs in poor nutrients. The authors also offer an interesting evolutionary explanation for this phenomenon.

The paper is well written, the experiments are well thought and the data supports the conclusions well. I particularly enjoyed the clarity of the discussion. This study describes an interesting phenomenon and provides plausible hypothesis to further investigate the molecular mechanisms responsible for it.

Minor comments:

1 - Fig. 1D should be moved to the supplementary Figure 1.

We would be happy to move Fig. 1D to Supplementary Fig. 1 as suggested.

2 - Writing about Fig. 1F, the authors state that "the histone H2B protein concentration decreases in inverse proportion with cell volume". The Fig. 1F data does not strongly support a proportional relationship between the variables, for that the exponent on the fitted regression should be -1 whereas it is closer to -1.5 on the graph. It is striking for instance that the YPGal population has a H2B expression more than twice the YPD population despite cells only being less than 25% less voluminous than in the YPD population and having the same cell-cycle phases distribution. Is there a technical bias in the H2B expression quantification or do the authors think that the relationship between the two variables is not strictly proportional?

We agree with the reviewer that the result of the quantification of H2B expression in YPGal is rather surprising especially because our flow cytometry analysis showed constant H2B levels across the different nutrient conditions. In fact, as shown in Fig. R1, we observe a relationship much closer to a power-law with exponent -1 if we exclude the YPGal data point. However, we don't have any specific reason to exclude the data point, for example for technical reasons, and we would therefore choose to still include it.

3 - The y-axis label of Fig. 1G should read "Actin" instead of "H2B".

We thank the referee for spotting this erroneous axis label, which we will correct accordingly.

4 - On Fig. 2A, the mRNA concentrations for HTB1 and HTB2 are lower in SCD than in YPD, however on Fig. 3B the mRNA concentration under the same promoters are similar in both media (if not slightly higher in SCD). Does that imply that, at least in SCD vs YPD, the promoter regulation is not sufficient to explain the differences in mRNA concentrations of HTB1 and HTB2?

We agree with the reviewer that this suggests that the promoter is not sufficient to capture the difference between SCD and YPD. Interestingly, we see in the smFISH experiments that in S-phase, the number of mRNA spots is lower in SCD compared to YPD. However, average concentrations as measured with qPCR also depend on expression outside S-phase, cell cycle fractions, and cell volumes, which can lead to complex effects. We will discuss this in the revised version of the manuscript.

Dr. Kurt M. Schmoller
Helmholtz Zentrum München
Institute of Functional Epigenetics
Ingolstädter Landstraße
Neuherberg 85764
Germany

9th Apr 2024

Re: EMBOJ-2024-116892

Decoupled transcript and protein concentrations ensure histone homeostasis in different nutrients

Dear Kurt,

Thank you again for your manuscript submission, and for now providing your tentative responses and revision plans. With some delay due to the Easter holidays, I have now had a chance to carefully go through them. I appreciate that your responses clarify many of the referees' concerns, and that your plans for further experimentation appear promising for deepening the mechanistic understanding of the observed phenomena. In this light, I shall be happy to invite you to revise the manuscript for The EMBO Journal, along the lines proposed. Regarding the requests for including additional approaches such as polysome profiling or bulk mass-spectrometry, I agree that based on the combined evidence from complementary experiments, this would not be essential within the scope of this revision.

I should remind you that it is our policy to allow only a single round of (major) revision, and I would therefore encourage you to update me should there be any unexpected problems with the revisions, or should you require an extension beyond the default three-months deadline. As always, competing manuscript published during the course of this revision will not affect our final decision on your study. Finally, please note the detailed information and guidelines on how to prepare a revision below and in our online Guide to Authors (particularly re. referencing, presentation of data in main figures and beyond, uploading of individual files etc.) - closely adhering to them shall greatly facilitate the editorial process at the time of resubmission.

Thank you again for the opportunity to consider this work, and I look forward to receiving your revision in due time.

With kind regards,

Hartmut

9) Digital image enhancement is acceptable practice, as long as it accurately represents the original data and conforms to community standards. If a figure has been subjected to significant electronic manipulation, this must be clearly noted in the figure legend and/or the 'Materials and Methods' section. The editors reserve the right to request original versions of figures and the original images that were used to assemble the figure. Finally, we generally encourage uploading of numerical as well as gel/blot image source data; for details see: embopress.org/page/journal/14602075/authorguide#sourcedata

At EMBO Press, we ask authors to provide source data for the main manuscript figures. Our source data coordinator will contact you to discuss which figure panels we would need source data for and will also provide you with helpful tips on how to upload and organize the files.

In the interest of ensuring the conceptual advance provided by the work, we recommend submitting a revision within 3 months (8th Jul 2024). Please discuss the revision progress ahead of this time with the editor if you require more time to complete the revisions. Use the link below to submit your revision:

Link Not Available

Referee #1 (Report for Author)

This is an interesting paper on an intriguing topic: how histone levels are kept constant (because they have to bind the same amount of DNA) in different nutrients. Unfortunately, it falls short of offering a convincing explanation. After their initial observations, they go on to nicely summarize their results for the constancy of histone levels in different nutrients as follows: "The apparent decoupling of histone protein and mRNA abundance suggests that medium-specific regulation of protein translation or stability is required to ultimately ensure histone homeostasis in different nutritional environments." However, they devote the rest of the paper to evaluating histone transcription in various settings instead of offering evidence about how the levels of these proteins are adjusted to where they need to be. This is bizarre and totally anticlimactic, detracting from the paper's significance. Every reader would like to know how histone protein amounts are maintained across nutrient conditions. That is the critical point, in my opinion. Not how histone mRNAs are downregulated in particular nutrients. Even their model in Figure 7 highlights "nutrient-specific translation" as the driver of the constant histone levels. Yet, their experiments do not focus on translation.

We agree with the reviewer that understanding the regulation of histone translation is now the big open question, and in the revised manuscript we performed new experiments that revealed a role of Tor1 in this regulation. However, we want to point out that many of the most important findings of our manuscript follow only after the quoted statement. In particular, we only afterwards showed that it is translation rather than protein degradation that is responsible for the decoupling, and that the decoupling is mediated by the untranscribed part of the promoter. These are critical insights towards understanding the decoupling, and we think that without this part of the paper, the obvious hypothesis would have been that the decoupling is simply mediated by protein degradation.

ADDITIONAL POINTS

- Despite their claim that "our data suggest that to compensate for the decreased transcript concentrations in poor nutrients, the relative translation efficiency of core histones is higher than in rich nutrients" (earlier in the text and the Discussion), they do not ever measure the translational efficiency of these mRNAs in different nutrients or the stability of the proteins. They rely exclusively on the single-cell fluorescence measurements of tagged proteins as a proxy. However, this does not measure translational efficiency. For any claims about translational efficiency, they would need to measure the ribosome-bound histone mRNAs and compare it to their total mRNA abundance. These are straightforward experiments. At a minimum, they should be performed to account for their observations. Ideally, a mechanism to explain the upregulation at the protein level should also be offered.

First, we would like to point out that in fact we measured protein stability (in the revised manuscript shown in Fig. EV5A-B), and we found that protein degradation does not compensate for the downregulation of histone transcripts in poor nutrients. These experiments, together with the single-cell measurements of mCitrine driven by the H2B

promoters (Fig. 5A-C), leave translation as the only possible process that ensures constant histone protein amounts despite the lower transcript amounts, and we therefore think our data support our claims.

While we do not think that measurement of the ribosome-bound fraction of histone mRNA is necessary for the claims we made in our manuscript and are beyond the scope of this manuscript, we agree with the reviewer that such experiments might provide interesting insights, and we are planning to go in this direction in the future.

- In contrast to their statements that cells in poor medium grow slower and are smaller, their data shows a complex relation between doubling time and cell size (compare Figure 1A with 1B). For example, YPGal cells grow faster than YPGE cells but are smaller, (with several other such comparisons). Likewise, although SDGE cells have a >2-fold longer doubling time than YPGE cells, their %G1 is the same or lower than YPGE cells. How do they explain these observations?

The reviewer is right that while it is a commonly stated trend that poor growth media lead to smaller cell sizes, in reality the relationship is complex. We have now clarified this in the revised manuscript. Importantly, we measure growth rate and cell size in each condition, and our conclusions do not depend on a dependence of cell size on growth rate.

- In Figure 1D, they argue that cells in poor nutrients have less protein from Ponceau-stained gels. The quantitative nature of these measurements is highly questionable (also, there are no error bars for the reference YPD samples in Figure 1E), but even with that assay, is there any difference among SCD, SCGE, SCGal, and YPGal, despite their different growth rates?

The error bars in the previous Fig. 1E (new Fig. 1D) show standard deviations between replicates. Because absolute quantification is not directly comparable between replicates, we first normalized on YPD for each individual replicate. By definition, YPD then shows the value 1, without variability between replicates that could be shown as standard deviation. While specific regulation depending on the exact media is not unlikely, we observed the expected general trend that large cells contain more protein. In the revised manuscript, we have included plots showing the relationship between cell size and doubling time (Fig. R1A, Fig. EV1A) and total protein and cell size (Fig. R1B, Fig. EV1C) to address this point and the one raised above. Our results indicate that the doubling time tends to decrease with increasing cell volume in the different growth media. At the same time, larger cells contain more total protein.

Figure R1. Population doubling times (A) and total protein content normalized on YPD (B) are as a function of cell volume in different nutrient conditions. Mean and standard deviation of $n = 4 - 6$ replicates are shown; line shows linear fit. This data are now shown in Fig. EV1.

- Comparing Figures 1F (relative H2B levels) to Figure 1G (relative Actin levels), they derive a central conclusion that "the histone H2B protein concentration decreases in inverse proportion with cell volume". However, much of the purported difference arises from a single outlier data point (YPGal). If one ignores that data point, the two graphs look very similar. This, along with the Ponceau approach to quantifying proteins, and immunoblots for H2B and actin, weakens these conclusions. A different, more quantitative method (histones and actin are abundant; why not use mass spectrometry-based methods?) should be added to the analysis.

In response to the reviewer's remark, we performed a new analysis without the data obtained for YPGal to assess the impact of this apparent outlier on the observed trend. As shown in Fig. R2, even without YPGal, our results support an inverse relationship between H2B protein concentration and cell volume. Interestingly, without the outlier, the relationship is even closer to the inverse proportionality that is expected for constant histone amounts.

Figure R2. Western blot analysis reveals that histone protein concentrations decrease with cell volume across nutrient conditions. Mean and standard deviation of at least 3 biological replicates are shown.

Still, we agree with the reviewer that western blots are not the most quantitative method. This is why we verified our finding with flow cytometry and fluorescence microscopy. Importantly, both these single cell methods also allow us to specifically measure histone amounts of G1 cells. This is important due to the cell cycle dependence of histone expression, which implies that in bulk experiments, the measured histone amounts also depend on the cell cycle distribution. All our measurements consistently – and following the obvious expectation – show that histone amounts at a given cell cycle stage are constant. We therefore do not think that bulk mass spectrometry experiments are necessary to make this point.

- Their fluorescence experiments in Figure 1H-K are better and support the conclusion that "histone protein amounts are maintained constant across nutrient conditions." However, the way the data is presented, with emphasis on Figure 1J, showing the three nutrients they examined and histone levels in the cell cycle related to cell volume at birth, confuses the issue. Even the section title "In different growth conditions, histone protein concentrations decrease with cell volume on a single cell level" distracts and confuses unnecessarily.

We adjusted the title of this section and aimed to clarify the relationship between decreasing histone concentrations with increasing volume and the fact that histone amounts are maintained constant.

- The cell cycle expression patterns in Figure 2 were obtained from an arrest (in M) and release. Then, they observe lower peak expression in the subsequent S phase in poor nutrients. The problem is that the cells continue to grow during the arrest, possibly confounding the outcome. They should have followed the second cycle to exclude arrest-related artefacts.

To exclude a possible effect of the nutrient-dependent cell volume on histone mRNA expression after cell cycle release, we compared the corresponding cell volumes at peak

expression. As shown in Figure R3 and the new Fig. EV2I, cells are larger in YPD than SCGE. Thus, the lower peak expression observed in SCGE is not a result of relatively increased cell volume.

Figure R3. Coulter counter measurements of cell volume at peak expression in YPD and SCGE. Lines indicate the mean across $n = 3$ replicate measurements, each shown as an individual dot.

Referee #2 (Report for Author)

In this manuscript, the authors investigate mechanisms of histone protein homeostasis in yeast cells growing under various nutrient conditions. A key finding is that histone mRNA expression changes in these different conditions, whereas protein levels do not change. They test hypotheses for how this regulation could work and conclude that the translation efficiency of the histone mRNA is increased in nutrient-rich conditions. Overall, the experiments are carefully done and the results are interesting. Although they don't identify factors or mRNA features that influence translational efficiency, these experiments will lay a foundation for future work. I support publication once the following issues have been addressed:

1. In the second paragraph of the Results section, the authors state: "we observe a decrease in total protein abundance in nutrient-poor conditions". Many readers will require more explanation here. As written, this could be interpreted to mean that protein concentration within cells decreases. However, since the same number of cells were used the decrease in protein abundance likely reflects the fact that cells are smaller in nutrient-poor conditions, but with similar protein concentration. It is also stated in this paragraph that "the histone H2B protein concentration decreases in inverse proportion with cell volume". For many readers, it may be more simple to state that the amount of histone protein per cell stays constant across different nutrient conditions that influence cell size.

We would like to thank the reviewer for these valid points. We agree that total protein content decreases in nutrient-poor conditions according to the smaller cell size, likely to achieve

roughly constant protein concentrations. At the same time, histone protein concentration decreases with increasing cell size in rich conditions, as histone proteins are maintained at constant amounts per cell. To avoid confusion, we have made the appropriate adjustments and clarifications in the revised manuscript.

2. In Figure 2A the authors use qPCR to quantify histone mRNA levels in different media. Using qPCR in these kinds of situations can be challenging because the results can depend strongly on the reference RNA used for normalization, which could be undergoing its own changes in the different conditions that would strongly influence the results. In Figure 2A, they use total RNA as the reference. Since the vast majority of total RNA is ribosomal RNA, they are essentially normalizing to rRNA. In Figure 2C they use *RDN18* to normalize. I didn't know what this is but looked it up and found that it is one of the ribosomal RNAs. The authors should edit to state that they are normalizing to rRNA in both cases. Importantly, they should include a comment or new data to describe how rRNA levels change in response to the different media, which will help readers interpret the data. This kind of information about the RNAs used for normalization should be included throughout wherever it is possible, and the authors should take extra care and explain their efforts to be sure that differences in histone mRNA levels between different media are not a qPCR artifact.

The reviewer is correct. Our reasoning was that rRNA constitutes the majority of total RNA. Thus, normalization on rRNA effectively mimics normalization on total RNA. In the revised version of the manuscript, we have clarified that we normalize to rRNA and included evidence that the ratio of *RDN18* RNA to total RNA is largely constant between the media (Fig. R4 and new Fig. EV2A).

Figure R4. C_q values of the reference gene *RDN18* obtained by RT-qPCR analysis across different nutrient conditions. Lines and error bars represent the means and standard deviations of $n = 6-8$ measurements.

3. The authors use a *sml1 Δ rad53 Δ spt21 Δ* strain investigate the effects of histone overexpression and conclude that cells in nutrient-poor conditions are more vulnerable to histone overexpression, based on analysis of the growth rate of strains in rich or poor media. However, the deleted genes have many other functions in the cells, and likely have additional unknown functions that have not yet been discovered. Thus, it is difficult to conclude that the effects of the mutants are due solely to effects on histone expression. The authors should acknowledge this caveat and use

appropriately cautious wording. It seems that a more rigorous approach would be to increase the copy number of histone genes. Has this been tried? Is there a mechanism that reduces histone protein levels in response to increased copy number? It would be helpful if the authors discussed these issues.

Histone homeostasis depends on multiple layers of regulation. In particular, a subset of genes shows transcriptional dosage compensation, and excess histone proteins are degraded in a Rad53-dependent manner (Eriksson et al., 2012). This is why instead of simply integrating additional gene copies, we chose the strategy of deleting Rad53 – which perturbs both the transcriptional and protein-degradation based control.

The reviewer is right that especially *rad53* deletion will have additional effects besides histone overexpression. In our manuscript, we followed the reasoning of Bruhn et al. that the fact that the phenotype is (partially) rescued by additional deletion of Spt21 points to a histone-specific effect. However, we agree with the reviewer, and now discuss this point more explicitly in the revised version of the manuscript.

4. The mechanism by which histone protein levels are adjusted in response to reduced transcript levels in poor carbon remain unknown. The author's experiments suggest that the 5' UTR is not required. They conclude that the translation efficiency of histone mRNAs is increased in nutrient-rich conditions, and that an "imprinting" mechanism is used to add factors that control mRNA translation during the transcription process. These are interesting ideas, although not a lot of data to support them, so the authors should use cautious wording that includes a thorough discussion of where the gaps in knowledge are.

We agree with the reviewer that further research is needed to unravel a potential imprinting mechanism underlying nutrient-dependent histone homeostasis. To gain new mechanistic insight, we have now investigated the impact of four carefully selected factors, Spt21, Hir1, Tor1 and Gcn2, on the regulation of histones across nutrients. Spt21 and Hir1 are involved in the activation and repression of histone gene regulation, respectively, by interacting with histone promoters. Tor1 and Gcn2 coordinate protein synthesis in response to nutrient availability. Using RT-qPCR and flow cytometry, we examined how cells lacking one of these four factors regulate histone mRNA and protein levels in rich and poor medium. Overall, our findings show that while Spt21 affects histone protein abundance in rich and poor conditions, Tor1 and Hir1 are critical for achieving the correct regulation of histone transcripts and maintaining constant amounts of histone proteins across nutrients. We have now included these new findings in the revised manuscript and extended the discussion accordingly.

Signed review: Doug Kellogg

Referee #3 (Report for Author)

Building on previous work having shown that unlike most other proteins, histone concentration sub-scales with cell volume to maintain constant histone-to-DNA levels, the authors show that volume-independent de-coupled transcription and translation of

histone genes contribute to homeostasis in different nutrient conditions. The authors show that, surprisingly, histone mRNA concentration is lower in poor nutrients due to regulation at the promoter level, suggesting that an increase in translation efficiency occurs in poor nutrients. The authors also offer an interesting evolutionary explanation for this phenomenon.

The paper is well written, the experiments are well thought and the data supports the conclusions well. I particularly enjoyed the clarity of the discussion. This study describes an interesting phenomenon and provides plausible hypothesis to further investigate the molecular mechanisms responsible for it.

We thank the reviewer for the kind words and positive feedback.

Minor comments:

1 - Fig. 1D should be moved to the supplementary Figure 1.

Following the reviewer's suggestion, we have moved Fig. 1D to Fig. EV1.

2 - Writing about Fig. 1F, the authors state that "the histone H2B protein concentration decreases in inverse proportion with cell volume". The Fig. 1F data does not strongly support a proportional relationship between the variables, for that the exponent on the fitted regression should be -1 whereas it is closer to -1.5 on the graph. It is striking for instance that the YPGal population has a H2B expression more than twice the YPD population despite cells only being less than 25% less voluminous than in the YPD population and having the same cell-cycle phases distribution. Is there a technical bias in the H2B expression quantification or do the authors think that the relationship between the two variables is not strictly proportional?

We agree with the reviewer that the result of the quantification of H2B expression in YPGal is rather surprising especially because our flow cytometry analysis showed constant H2B levels across the different nutrient conditions. In fact, as shown in Fig. R2, we observe a relationship much closer to a power-law with exponent -1 if we exclude the YPGal data point. However, we don't have any specific reason to exclude the data point, for example for technical reasons, and we would therefore choose to still include it.

3 - The y-axis label of Fig. 1G should read "Actin" instead of "H2B".

We thank the reviewer for spotting this erroneous axis label, which we have corrected accordingly.

4 - On Fig. 2A, the mRNA concentrations for HTB1 and HTB2 are lower in SCD than in YPD, however on Fig. 3B the mRNA concentration under the same promoters are similar in both media (if not slightly higher in SCD). Does that imply that, at least in SCD vs YPD, the promoter regulation is not sufficient to explain the differences in mRNA concentrations of HTB1 and HTB2?

We agree with the reviewer that this suggests that the promoter is not sufficient to capture the difference between SCD and YPD. Interestingly, we see in the smFISH experiments that

in S-phase, the number of mRNA spots is lower in SCD compared to YPD. However, average concentrations as measured with qPCR also depend on expression outside S-phase, cell cycle fractions, and cell volumes, which can lead to complex effects. In the revised manuscript, we now discuss that the promoter alone does not account for the differences between SCD and YPD medium observed for histone genes.

Dr. Kurt M. Schmoller
Helmholtz Zentrum München
Institute of Functional Epigenetics
Ingolstädter Landstraße
Neuherberg 85764
Germany

10th Jul 2024

Re: EMBOJ-2024-116892R

Decoupled transcript and protein concentrations ensure histone homeostasis in different nutrients

Dear Kurt,

Thank you for submitting your revised manuscript for our consideration. It has now been re-reviewed, and both referees 2 and 3 were fully satisfied with the revisions. Referee 1 still insists on certain additional experiments, which based on our earlier discussions as well as input from the other referees we had however decided not to require as prerequisite for publication of this work. Following incorporation of the following editorial issues, we shall therefore be happy to accept the study for The EMBO Journal:

- Please upload all main Figures and all Expanded View (EV) figures as individual files, with sufficient resolution/quality for production. Main and EV figure legends should be collated at the end of the manuscript text
- Please collate all Appendix Figures, together with their respective legend below each figure, in a dedicated PDF called "Appendix", and headed by a brief Table of Contents listing the article title and the contents with page numbers. Please make sure to correctly name these figures "Appendix Figure S1/2/3..." both within the Appendix and when referencing them in the main text.
- On the abstract page of the manuscript, please include 4-5 general keyword terms to enhance searchability.
- Please adjust the order of the manuscript sections: Title page with complete author information, Abstract, Keywords, Introduction, Results, Discussion, Materials & Methods, Data Availability Section, Acknowledgements, Disclosure and Competing Interests Statement, References, Main figure legends, Tables, Expanded Figure Legends.
- Please adjust the format of the reference list and of the in-text citations according to EMBO Journal format (alphabetical order rather than numbered, author name et al + year...). Also, please adjust the format for citation of preprints as specified in our author guidelines:
The citation in the text should be: "(preprint: NAME1 et al, YEAR)"; The citation in the reference list: "Author NAME1, Author NAME2, ... (YEAR) article title. bioRxiv doi: XXX"
- Please include a Disclosure and competing interests statement (next to the Acknowledgment section) - for details, see <https://www.embopress.org/competing-interests>
- Please double-check to make sure to all relevant funding information in the manuscript is congruent with the info entered into our submission system (e.g. "the Helmholtz Association" may need to be entered as a funder in eJP).
- Please do complete the Source Data checklist we sent you at the early revision stage (attached once more), and do use the free-text box to clarify where relevant Source Data not uploaded with the revised manuscript can be found and accessed (repository plus accession codes). Note that Source Data needs to be submitted to the journal with the revised version, OR hosted in an official public repository, rather than on an internal server.
- For Data and Code Availability sections, please refer to the respective section of our Guide to Authors on how to present this information (<https://www.embopress.org/page/journal/14602075/authorguide#dataavailability>).
- Please make sure to call out each subpanel for Figure 7 (e.g., "Figure 7A-C" instead of "Figure 7")
- During routine pre-acceptance checks, our data editors have raised the following queries regarding figures, data, and legends, which I would ask you to address (ideally using the Track Changes option):
 1. Please note that the exact p values are not provided in the legends of figures 2a; 3b; 4a, c; 5b-c, e, g-h; EV 1e; EV 2k; EV 3g-h.
 2. Please note that in figures 2a; 3b; there is a mismatch between the annotated p values in the figure legend and the annotated

p values in the figure file that should be corrected.

3. Please note that information related to "n" is missing in the legends of figures 1j; EV 4a-j.

4. Please note that the measure of center for the error bars needs to be defined in the legends of figures 1c-d; 4c-e; 6b-c.

- Please enter valid email addresses for all coauthors in the submission system, so that they could be informed about the submission and final decision. At resubmission, acknowledgement emails failed to be delivered to Dimitra Chatzitheodoridou - dimitra.chatzitheodoridou@helmholtz-munich.de

- Finally, please provide suggestions for a short 'blurb' text prefacing and summing up the study in two sentences (max. 250 characters), followed by 3-5 one-sentence 'bullet points' with brief factual statements of key results of the paper; they will form the basis of an editor-written 'Synopsis' accompanying the online version of the article. Please also upload a synopsis image, which can be used as a "visual title" for the synopsis section of your paper (maybe based on a more compact version of Figure 7?). The image should be in PNG or JPG format with the modest dimensions of EXACTLY 550 pixels wide and 300-600 pixels high.

I am therefore returning the manuscript to you for a final round of revision, to allow you to make these modifications and upload the revised files - ideally together with an itemized reply to the above editorial request, to facilitate our checking. Once we will have received them, we should hopefully be ready to swiftly proceed with formal acceptance and production of the manuscript.

With kind regards,

Hartmut

8) Please note that supplementary information at EMBO Press has been superseded by the 'Expanded View' for inclusion of

additional figures, tables, movies or datasets; with up to five EV Figures being typeset and directly accessible in the HTML version of the article. For details and guidance, please refer to:
embopress.org/page/journal/14602075/authorguide#expandedview

9) Digital image enhancement is acceptable practice, as long as it accurately represents the original data and conforms to community standards. If a figure has been subjected to significant electronic manipulation, this must be clearly noted in the figure legend and/or the 'Materials and Methods' section. The editors reserve the right to request original versions of figures and the original images that were used to assemble the figure. Finally, we generally encourage uploading of numerical as well as gel/blot image source data; for details see: embopress.org/page/journal/14602075/authorguide#sourcedata

At EMBO Press, we ask authors to provide source data for the main manuscript figures. Our source data coordinator will contact you to discuss which figure panels we would need source data for and will also provide you with helpful tips on how to upload and organize the files.

In the interest of ensuring the conceptual advance provided by the work, we recommend submitting a revision within 3 months (8th Oct 2024). Please discuss the revision progress ahead of this time with the editor if you require more time to complete the revisions. Use the link below to submit your revision:

Link Not Available

Referee #1:

The authors' responses was dissapointing. They argue for a role in translation 'by exclusion', without directly testing their main conclusion. The test is simple. Growing the cells in different conditions, isolating ribosomes, and quantifying histone transcripts bound to ribosomes vs. total mRNA levels by qPCR is quick (a matter of a few days), reasonable, and easy. Instead of directly addressing my main concern experimentally, as outlined above, they make circuitous and largely irrelevant arguments. Regrettably, I cannot support the publication of this work in the EMBO Journal.

Referee #2:

The authors have adequately addressed each of the concerns that were raised in my previous review.

Referee #3:

My general opinion on the paper hasn't changed since the initial version. The authors have addressed my comments.

All editorial and formatting issues were resolved by the authors.

Dr. Kurt M. Schmoller
Helmholtz Zentrum München
Institute of Functional Epigenetics
Ingolstädter Landstraße
Neuherberg 85764
Germany

27th Aug 2024

Re: EMBOJ-2024-116892R1

Decoupled transcript and protein concentrations ensure histone homeostasis in different nutrients

Dear Kurt,

Thank you for submitting your final revised manuscript for our consideration. I am pleased to inform you that we have now accepted it for publication in The EMBO Journal.

With kind regards,

Hartmut
